# DRAW: Domain Weight Randomization with Bayesian Updating for LLM Pre-Training

**Ruonan Wang**                                                    *wangrn117@nenu.edu.cn*
*School of Mathematics and Statistics*
*Northeast Normal University*

**Yongqi Qiao**                                              *miracleqiao2430@gmail.com*
*Academy for Advanced Interdisciplinary Studies*
*Peking University*

**Zhonglin Xie**                                                       *zlxie@pku.edu.cn*
*Beijing International Center for Mathematical Research*
*Peking University*

**Kun Yuan**$^*$                                                      *kunyuan@pku.edu.cn*
*Center for Machine Learning Research*
*Peking University*
*AI for Science Institute, Beijing, China*

**Reviewed on OpenReview:** *https://openreview.net/forum?id=tc8TyD7ZyD*

## Abstract

Optimal pre-training data mixture is pivotal for large language model (LLM) performance, but searching for the best domain weights is computationally expensive. We present Domain Weight Randomization with Bayesian Updating (DRAW), a principled framework treating domain weights as Dirichlet-distributed random variables whose parameters scale with model width. Informative priors are first estimated using proxy models; the main model then refines these using Bayesian inference and parameter scaling, dynamically sampling domain weights during training. Theoretically, DRAW reduces generalization error at a rate $\mathcal{O}(1/\sqrt{n})$ as model width increases, ensuring stable convergence. Empirical results on open-domain corpora and diverse benchmarks show DRAW reliably outperforms fixed and adaptive baselines in both language modeling and downstream tasks, achieving better average and worst-case performance alongside strong robustness. DRAW not only highlights valuable data domains while suppressing noisy ones, but also introduces a scalable and effective mechanism for adaptive data mixing in LLM pre-training, facilitating efficient knowledge transfer from proxy to large models.

## 1 Introduction

The proportion of data sourced from each domain is increasingly recognized as a crucial factor shaping the quality and generalization capability of large language models (LLMs) during pre-training (Brown et al., 2020; Grattafiori et al., 2024; Zhou et al., 2025). Recent studies demonstrate that careful adjustment of these mixture ratios can have a direct and significant impact on downstream task performance and model robustness, particularly as the scale and diversity of training data expand (Gu et al., 2025; McKinzie et al., 2024; Grattafiori et al., 2024). While both heuristic (Gao et al., 2020; Shen et al., 2023) and optimization-based (Xie et al., 2023; Fan et al., 2024; Liu et al., 2025) approaches have been proposed to determine optimal mixture ratios, substantial challenges remain. As the number of data domains and model size grow, existing

---

$^*$Corresponding author. Email: kunyuan@pku.edu.cn

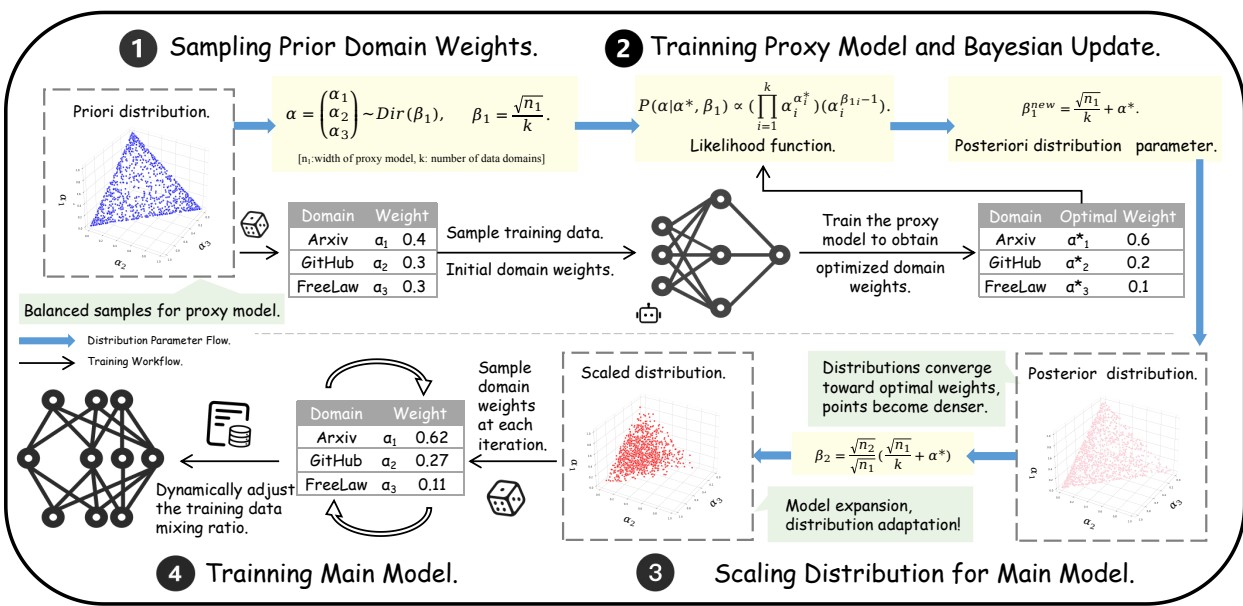

Figure 1: Overview of the proposed workflow, illustrated with `Arxiv`, `GitHub`, and `FreeLaw` as example domains. First, based on the width of the proxy model and the number of data domains, we determine the parameters of the Dirichlet prior distribution and sample a set of domain weights as the initial weights for the proxy model (**Step 1**). Next, we train the proxy model to obtain optimized domain weights, and perform a Bayesian update to estimate the posterior distribution parameters (**Step 2**). The resulting posterior distribution is then scaled by a factor determined by the ratio of the widths of the main and proxy models, enabling migration of the learned distribution to the main model (**Step 3**). Finally, during the training of the main model, we sample a set of domain weights from the final distribution at each iteration, use these weights to sample data, and thus dynamically train the main model (**Step 4**).

methods are challenged by inefficiency and limited scalability, underscoring the need for more principled frameworks for data mixture adaptation.

A common approach to transferring data mixture policies across models assumes *rank invariance*—that is, the optimal mixture ratios determined on a small proxy model can be directly applied to larger models (Liu et al., 2025). However, this assumption is often violated in practice, as large language models tend to exhibit greater sensitivity to the composition of training data: even slight variations in domain proportions may cause substantial performance fluctuations, particularly for rare domains (Grattafiori et al., 2024). However, there are also studies that have treated domain weights as adaptive parameters, learning their optimal values either through model scaling or jointly in a specific data environment (Ye et al., 2025; Shukor et al., 2025; Ge et al., 2025). Such methods adjust domain mixtures based on scaling laws or model-specific factors, thus alleviating some of the limitations of naive weight transfer and resulting in more robust and generalizable LLMs as both model and data scales increase. Nevertheless, it is important to note that most existing approaches maintain fixed data mixture ratios throughout the entire training process, and do not allow the mixture to evolve dynamically as training progresses or the model grows.

To address this, we propose the **D**omain **W**eight **Ra**ndomization Model (DRAW). Instead of treating domain weights as fixed constants, DRAW models them as random variables drawn from distributions whose parameters are tied to model size. This reframes proxy model training from finding one set of optimal weights to dynamically updating parameters of a weight distribution via a Bayesian framework (van de Schoot et al., 2021). We further introduce a scaling factor to adapt the parameterized distribution to the main model. At each pre-training step, domain weights are sampled from this distribution to improve adaptability and generalization. DRAW consists of four primary steps (Figure 1): (1) sampling the initial weight distribution based on the proxy model width; (2) training the proxy to obtain optimized weights; (3) utilizing Bayesian updat-

ing and scaling to adjust the distribution for the main model; (4) during main model training, dynamically sampling weights to guide data mixing, enabling robust adaptation.

Our experiments demonstrate that DRAW substantially improves the adaptability and generalization of large-scale model pre-training, outperforming mainstream alternatives in both average and worst-case validation loss , and surpassing them on several downstream tasks (e.g., MMLU (Hendrycks et al., 2021), GSM8K (Cobbe et al., 2021)). The method achieves effective weight stratification and sparsification—focusing on key domains while suppressing noise and irrelevant data—and both theoretical analysis and empirical results confirm reduced parameter estimation errors and stable training dynamics. Furthermore, DRAW exhibits remarkable robustness to the choice of proxy model size and random initialization, consistently delivering state-of-the-art(SOTA) results with minimal performance variance across seeds. These findings validate the scalability and reliability of the proposed framework for adaptive domain weighting in large language model pre-training.

Our main contributions are as follows:

- We propose a Dirichlet-based domain weight randomization method, achieving more principled and effective data mixture strategies for large language model pre-training.

- We develop a Bayesian updating and parameter scaling framework, which improves the stability and accuracy of domain weight transfer across model scales.

- We provide theoretical analysis showing that randomized domain weights and Bayesian adaptation guarantee stable convergence and reduce estimation error at scale.

- We experimentally show that DRAW consistently outperforms other mainstream methods, demonstrating its superior generality and robustness.

## 2 Related work

Our work builds on prior studies that explore how to design and adjust data mixing strategies for multi-domain pre-training, with a particular focus on transferring effective mixing policies learned from small-scale proxy models to large-scale foundation models. We identify three main research directions relevant to this goal: data mixing, scaling laws and randomization of domain weights.

**Scaling Laws.** The scaling laws for large models, proposed by OpenAI in 2020 (Kaplan et al., 2020). It describes a power-law relationship between training loss and factors such as computational resources, model parameterization, and training data size (Hoffmann et al., 2022). In 2023, Greg Yang et al. introduced Maximum Update Parameterization (MUP), which enables the optimal parameters determined on small proxy models to predict the loss of larger models through a modified power law (Yang & Hu, 2021).

The existing literature on data mixture scaling laws generally takes one of two perspectives. One approach assumes that the optimal domain weights learned by small-scale models can be directly transferred to large-scale models without modification (Xie et al., 2023; Fan et al., 2024; Liu et al., 2025). However, this static allocation overlooks the differences introduced by model scaling. Another approach fits domain weights as parameters according to scaling laws, yet this method struggles to handle high-dimensional settings with many data domains (Gu et al., 2024; Goyal et al., 2024; Béthune et al., 2025). To address these issues, we introduce a scaling factor that dynamically adjusts the domain weight distribution parameters when transferring from small models to large models. This enables effective adaptation of domain weights across different model sizes, thus enhancing the flexibility and generalization capability of large-scale language model pre-training.

**Weight Randomization and Data Augmentation.** Random parameter initialization is a common technique in deep learning used to enhance exploration and prevent problems such as vanishing or exploding gradients during training (Goodfellow et al., 2016). In the computer vision domain, data augmentation strategies such as mixup (Zhang et al., 2018), CutMix (Yun et al., 2019), and SnapMix (Huang et al., 2021) have shown remarkable effectiveness by mixing training data samples according to mixing coefficients

sampled from a Beta distribution. This process of stochastic data mixing not only increases the diversity of training examples, thus improving generalization, but also acts as a form of regularization to reduce overfitting. We extend these ideas beyond simple two-way mixes: for multi-domain data in language model pre-training, domain weights can be considered as random vectors on the simplex, naturally modeled by the Dirichlet distribution, a multidimensional generalization of the Beta distribution (Bae et al., 2024). This allows adaptive sampling of domain proportions when constructing each batch, enabling the model to exploit both the diversity and heterogeneous quality of different data sources during large-scale pre-training.

**Dynamic Refinement Methods.** DRAW is also related to recent methods (Jiang et al., 2024) that refine data or domain weights online during training based on model feedback. The key distinction is that DRAW separates *estimation* from *sampling*: a proxy model is first used to infer an informative domain prior, and the main model is then trained by stochastic sampling around this prior. In contrast, online refinement methods update weights directly from short-term training dynamics. This proxy-first design aims to retain the benefits of adaptive sampling while reducing sensitivity to transient minibatch fluctuations.

## 3 Method

### 3.1 DRAW

**Setup.** Suppose we have a dataset $D_{\text{train}} \triangleq \{D_1, \ldots, D_k\}$ consisting of a mixture of $k$ distinct domains, where each domain $D_i$ has an associated dataset. Each domain $D_i$ is assigned a domain weight $\alpha_i$, and together these weights define the data sampling distribution as $P_{\boldsymbol{\alpha}} = \sum_{i=1}^{k} \alpha_i \cdot \text{unif}(D_i)$, where $\text{unif}(D) = \frac{1}{|D|} \sum_{x \in D} \delta_x$ denotes the uniform distribution over dataset $D$, and $\delta_x(x')$ is 1 if $x' = x$ and 0 otherwise. The domain weight vector $\boldsymbol{\alpha} \triangleq \{\alpha_1, \ldots, \alpha_k\} \in \Delta^k$ is treated as a random variable drawn from a Dirichlet distribution.

We consider two Transformer-based language models that share the same architectural depth (i.e., number of layers) but differ in width: a smaller proxy model with width $n_1$, and a larger main model with width $n_2$. Here, width refers to the number of hidden units per layer, as is standard in prior work (Yang & Hu, 2021). For mainstream LLMs such as Qwen3 (Qwen Team, 2025), increasing model width expands the hidden state, the intermediate feedforward network (FFN) dimension, and the input/output embedding sizes, while keeping the number of layers fixed. The proxy model is first trained to optimize domain weight distributions suitable for its scale, and these learned distributions are then used to initialize the main model. The main model is subsequently trained to minimize the training loss on the pre-defined objectives, dynamically adapting data mixture via the domain weights informed by the proxy.

**Step 1: Sampling Prior Domain Weights.**

We begin by establishing a Dirichlet prior over the domain weights, with concentration parameters set as a function of both the proxy model width $n_1$ and the number of domains $k$. Specifically,

$$\boldsymbol{\beta_1} = \left( \frac{\sqrt{n_1}}{k}, \ \frac{\sqrt{n_1}}{k}, \ \ldots, \ \frac{\sqrt{n_1}}{k} \right). \tag{1}$$

Domain weights are then sampled as $\boldsymbol{\alpha} \sim \text{Dir}(\boldsymbol{\beta_1})$. Note that this parameterization does *not* change the mean of the Dirichlet distribution, which remains uniform over domains, i.e., $\mathbb{E}[\alpha_i] = 1/k$ for all $i$. Instead, increasing $n_1$ increases the *total concentration* of the prior, causing samples to cluster more tightly around the mean and reducing sampling variance. In other words, a larger proxy model induces a *sharper* and more confident prior over domain mixtures, rather than a more uniform one. This design reflects the intuition that stronger proxy models can provide more stable estimates of desirable data mixtures, while smaller proxies may benefit from broader stochastic exploration. The sampled prior then serves as a principled starting point for subsequent Bayesian refinement of the domain weights.

**Step 2: Training Proxy Model and Bayesian Updating.**

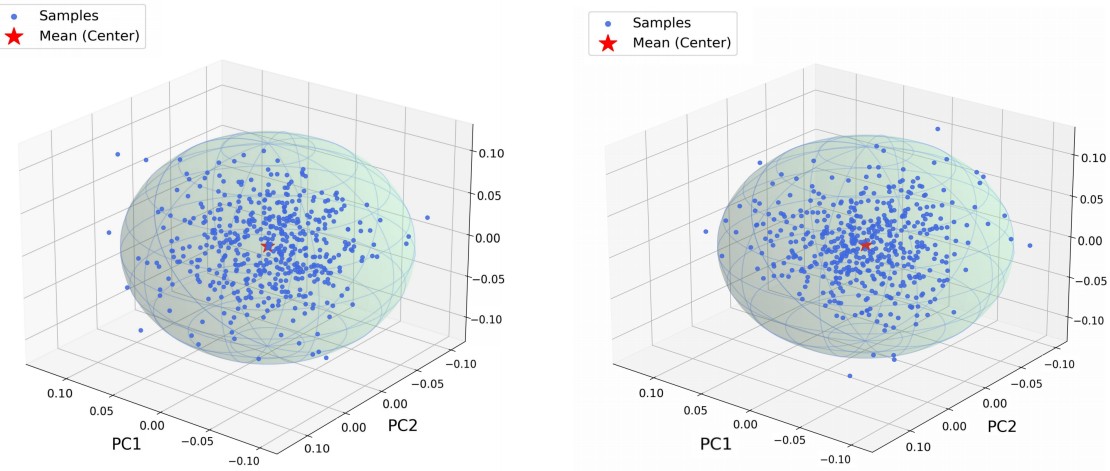

Figure 2: 3D visualization of domain weight samples over the 17 sub-domains in The Pile. **Left:** untrained random weights from a Dirichlet distribution with parameter $\frac{\sqrt{n_2}}{k}\mathbf{1}$. **Right:** learned weights of the main model after DRAW training.

To estimate the parameters governing the domain weight distribution, we combine a DRO-style proxy optimization step with Bayesian updating. At the proxy stage, we solve

$$\min_{\boldsymbol{\theta}} \max_{\boldsymbol{\alpha} \in \Delta^k} \mathcal{L}(\boldsymbol{\theta}, \boldsymbol{\alpha}) := \sum_{i=1}^{k} \alpha_i \frac{1}{|D_i|} \sum_{x \in D_i} \widetilde{\ell}(x; \boldsymbol{\theta}), \tag{2}$$

where $\widetilde{\ell}(x; \boldsymbol{\theta}) = \ell_{\boldsymbol{\theta}}(x) - \ell_{\text{ref}}(x)$ is the excess loss relative to a reference model (Xie et al., 2023). This formulation is inspired by Group DRO (Sagawa et al., 2020) robust reweighting, but in DRAW it is used only to infer a proxy-level domain prior. The resulting optimized weights $\boldsymbol{\alpha}^*$ are then treated as pseudo-counts for Bayesian updating, after which the main model is trained via stochastic Dirichlet sampling around the proxy-informed prior rather than by directly optimizing a fixed DRO mixture.

To refine the distribution, we perform a Bayesian update using $\boldsymbol{\alpha}^*$ as pseudo-observations, yielding a posterior Dirichlet distribution:

$$P(\boldsymbol{\alpha} \mid \boldsymbol{\alpha}^*, \boldsymbol{\beta}_1) \propto P(\boldsymbol{\alpha}^* \mid \boldsymbol{\alpha})\, P(\boldsymbol{\alpha} \mid \boldsymbol{\beta}_1) \propto \left[\prod_{i=1}^{k} \alpha_i^{\alpha_i^*}\right] \left[\prod_{i=1}^{k} \alpha_i^{\beta_{1i}-1}\right] = \prod_{i=1}^{k} \alpha_i^{\beta_{1i}+\alpha_i^*-1}. \tag{3}$$

which is a Dirichlet with updated concentration parameters $\boldsymbol{\beta}_1^{\text{new}} = \boldsymbol{\beta}_1 + \boldsymbol{\alpha}^*$. This posterior integrates both prior knowledge and observed domain importance, establishing a statistically principled basis for the mixing distribution used in the subsequent main model training.

**Step 3: Scaling Distribution for Main Model.**

Instead of directly applying the domain weight distribution optimized for the proxy model (width $n_1$), we scale the posterior Dirichlet parameters to account for increases in model capacity. Given the updated concentration vector $\boldsymbol{\beta}_1^{\text{new}}$, we define the main model's parameters as:

$$\boldsymbol{\beta}_2 = \frac{\sqrt{n_2}}{\sqrt{n_1}}\, \boldsymbol{\beta}_1^{\text{new}}, \tag{4}$$

where $\boldsymbol{\beta}_2$ specifies the Dirichlet distribution for the main model of width $n_2$. This scaling results in a sharper domain weighting distribution, i.e., lower-variance sampling around the proxy-informed mean mixture. Specifically, each Dirichlet parameter can be written as:

$$\boldsymbol{\beta}_2 = \frac{\sqrt{n_2}}{\sqrt{n_1}}\, \boldsymbol{\alpha}^* + \frac{\sqrt{n_2}}{k}\mathbf{1}, \tag{5}$$

where $\mathbf{1}$ is a $k$-dimensional vector of ones. This procedure transfers the proxy-estimated mixture to the main model in a scale-aware stochastic form.

**Remark 1.** *Although in Eq. (5) the uniform baseline term $\frac{\sqrt{n_2}}{k}\mathbf{1}$ may dominate in magnitude when $k$ is small or $n_1 \approx n_2$, this does not mean that the influence of $\boldsymbol{\alpha}^*$ is diminished. In a Dirichlet distribution, each concentration parameter affects the mean via*

$$\mathbb{E}[\alpha_i] = \frac{\frac{\sqrt{n_1}}{k} + \alpha_i^*}{\sqrt{n_1} + 1}.$$

*Thus, $\boldsymbol{\alpha}^*$ shifts the expected proportions away from the uniform distribution, even when its scale is numerically much smaller than the baseline term.*

*This design is intentional: a large baseline concentration ensures low-variance and stable domain weights, preventing overfitting to the fine-grained biases learned by a large proxy model, while $\boldsymbol{\alpha}^*$ still provides directional bias toward the proxy-learned structural preferences. As $n_1$ grows, the relative size of the $\boldsymbol{\alpha}^*$ term decreases, reflecting a robustness mechanism rather than a loss of useful information. Increasing the concentration of the Dirichlet distribution does not make the mixture more uniform; instead, it reduces sampling variance around the proxy-informed mean. As observed in Suriya et al (Gunasekar et al., 2023), this appears beneficial for larger main models, which benefit from a higher signal-to-noise ratio in domain sampling while still retaining controlled stochastic exploration.*

### Step 4: Training Main Model.

At each iteration of main model training, domain weights are dynamically sampled from the scaled Dirichlet prior $\mathrm{Dir}(\boldsymbol{\beta}_2)$ (see Figure 2), and data batches are drawn accordingly. As illustrated in Figure 2, the learned domain weight distribution after scaling is both more concentrated and noticeably non-uniform, with prominent biases towards certain domains. Such concentration enables the model to focus on more informative domains, thereby further enhancing the robustness and efficiency of the DRAW framework under heterogeneous data distributions.

## 3.2 Theoretical Guarantees

**Preliminaries and Notation.** Let $F(\boldsymbol{\theta})$ constitute the objective function parameterized by $\boldsymbol{\theta} \in \mathbb{R}^d$, and let $\|\cdot\|$ denote the Euclidean ($\ell_2$) norm. The following definitions are used:

- **Simplex ($\Delta_k$):** The probability simplex of dimension $k-1$ is denoted as $\Delta_k = \{\boldsymbol{\alpha} \in \mathbb{R}^k \mid \sum_{i=1}^k \alpha_i = 1, \alpha_i \geq 0\}$.

- **Strong Convexity:** $F$ is $\mu$-strongly convex if for any $\boldsymbol{\theta}, \boldsymbol{\theta}' \in \mathbb{R}^d$ and $\lambda \in [0, 1]$,

$$F(\lambda\boldsymbol{\theta} + (1-\lambda)\boldsymbol{\theta}') \leq \lambda F(\boldsymbol{\theta}) + (1-\lambda)F(\boldsymbol{\theta}') - \frac{\mu}{2}\lambda(1-\lambda)\|\boldsymbol{\theta} - \boldsymbol{\theta}'\|^2.$$

- **$L$-smoothness:** $F$ is $L$-smooth if it is differentiable and $\|\nabla F(\boldsymbol{\theta}) - \nabla F(\boldsymbol{\theta}')\| \leq L\|\boldsymbol{\theta} - \boldsymbol{\theta}'\|$ for all $\boldsymbol{\theta}, \boldsymbol{\theta}' \in \mathbb{R}^d$.

- **Convergence in Probability:** A sequence $\{\boldsymbol{\theta}_T\}$ converges in probability to $\boldsymbol{\theta}^*$, denoted as $\boldsymbol{\theta}_T \xrightarrow{p} \boldsymbol{\theta}^*$, if $\lim_{T\to\infty} \mathbb{P}(\|\boldsymbol{\theta}_T - \boldsymbol{\theta}^*\| > \epsilon) = 0$ for any $\epsilon > 0$.

- **Asymptotic Notation ($\Theta$):** For a quantity $x$ depending on a scaling parameter $n$ (e.g., width), $x = \Theta(1)$ implies that there exist constants $C_1, C_2 > 0$ such that $C_1 \leq |x| \leq C_2$ for sufficiently large $n$.

- **$M_1$-Lipschitz Continuity:** The loss function $L(\boldsymbol{\theta}, \boldsymbol{\alpha})$ is $M_1$-Lipschitz with respect to $\boldsymbol{\alpha}$ if $|L(\boldsymbol{\theta}, \boldsymbol{\alpha}) - L(\boldsymbol{\theta}, \boldsymbol{\alpha}')| \leq M_1\|\boldsymbol{\alpha} - \boldsymbol{\alpha}'\|$ for any $\boldsymbol{\alpha}, \boldsymbol{\alpha}' \in \Delta_k$.

We provide a rigorous theoretical analysis of DRAW under adaptive, randomized domain weights (see Appendix C for proofs). Specifically, we derive formal guarantees governing model output stability, optimization trajectory convergence, and generalization error scaling.

**Lemma 1.** *Based on the Tensor Program and Maximal Update Parametrization, the size of the model output $f$ after $t$ iterations is $\Theta(1)$.*

Lemma 1 demonstrates the inherent stability of the DRAW approach. By theoretically bounding the variance of the stochastic gradients induced by domain weight sampling, it ensures that the training process remains controlled, effectively mitigating risks of optimization divergence. This theoretical property is corroborated by the stable losses observed in Figure 5. Moreover, this result justifies our choice of Dirichlet hyperparameters, as they directly regulate the trade-off between sampling diversity and training stability.

**Theorem 1.** *Suppose $F(\boldsymbol{\theta}) = \mathbb{E}_{\boldsymbol{\alpha}}[\mathcal{L}(\boldsymbol{\theta}, \boldsymbol{\alpha})]$ is $\mu$-strongly convex and $L$-smooth. If the parameters $\boldsymbol{\theta}$ are updated using stochastic gradients computed with domain weights $\boldsymbol{\alpha}_t \sim \mathrm{Dir}(\boldsymbol{\beta})$ using a step size sequence $\{\eta_t\}$ satisfying $\sum \eta_t = \infty$ and $\sum \eta_t^2 < \infty$, then the optimization guarantees convergence in mean square to the unique minimizer $\boldsymbol{\theta}^*$ of $F(\boldsymbol{\theta})$:*

$$\lim_{T \to \infty} \mathbb{E}\left[\|\boldsymbol{\theta}_T - \boldsymbol{\theta}^*\|^2\right] = 0, \quad \implies \quad \boldsymbol{\theta}_T \xrightarrow{p} \boldsymbol{\theta}^*. \tag{6}$$

Theorem 1 shows that, under standard strong-convexity and smoothness assumptions, stochastic optimization with Dirichlet-sampled domain weights remains mean-square convergent to the minimizer of the *expected* objective $F(\boldsymbol{\theta})$. The result should be interpreted as a stability guarantee for randomized domain weighting: the additional stochasticity introduced by sampling $\boldsymbol{\alpha}_t$ does not, by itself, prevent convergence under a standard diminishing-step-size regime. Importantly, this theorem does not claim that practical large-scale pretraining is globally convex, nor that stochastic weighting alone guarantees better final model quality; rather, it formalizes that DRAW's randomized weighting mechanism is compatible with stable optimization in an idealized setting.

**Theorem 2.** *Let $\mathcal{L}(\boldsymbol{\theta}, \boldsymbol{\alpha})$ be the loss function of a model with width $n_2$ parameterized by $\boldsymbol{\theta}$, under domain mixture weights $\boldsymbol{\alpha}$. Assume the loss is $M_1$-Lipschitz regular with respect to $\boldsymbol{\alpha}$. Define the expected risk as:*

$$F(\boldsymbol{\theta}) := \mathbb{E}_{\boldsymbol{\alpha} \sim \mathrm{Dir}(\boldsymbol{\beta})}[\mathcal{L}(\boldsymbol{\theta}, \boldsymbol{\alpha})],$$

*where $\boldsymbol{\beta} = \frac{n_1}{k} + \boldsymbol{\alpha}^*$ (implying concentration parameter $n_1$). Let $\boldsymbol{\theta}^*$ denote the population minimizer and $\hat{\boldsymbol{\theta}}$ the parameter obtained after optimization.*

*The generalization error, governed by the interplay between domain variance (scale $n_1$) and model width (scale $n_2$), is bounded by:*

$$F(\hat{\boldsymbol{\theta}}) - F(\boldsymbol{\theta}^*) \leq M_1 \sqrt{\left(\sum_{i=1}^{k} \frac{\beta_i(B - \beta_i)}{B^2(n_2 B + n_1)}\right)},$$

*where $B = n_1 + 1$ is the sum of Dirichlet parameters.*

*In the regime of large data concentration ($n_1 \gg 1$) and wide models ($n_2 \gg 1$), the scaling law simplifies to:*

$$F(\hat{\boldsymbol{\theta}}) - F(\boldsymbol{\theta}^*) = O\left(\frac{M_1}{\sqrt{n_1 n_2}}\right).$$

**Remark 2.** *The theoretical bound specifically isolates the generalization error induced by the stochasticity of Dirichlet-sampled data mixing, independent of optimization noise (e.g., SGD randomness). Assuming the model reaches a parameter configuration close to the population minimizer, this upper bound primarily reflects the variance introduced by data mixture variability.*

Theorem 2 bounds the excess risk induced by stochastic variation in Dirichlet-sampled domain mixtures, and shows that this contribution decreases as either the proxy concentration scale $n_1$ or the main model

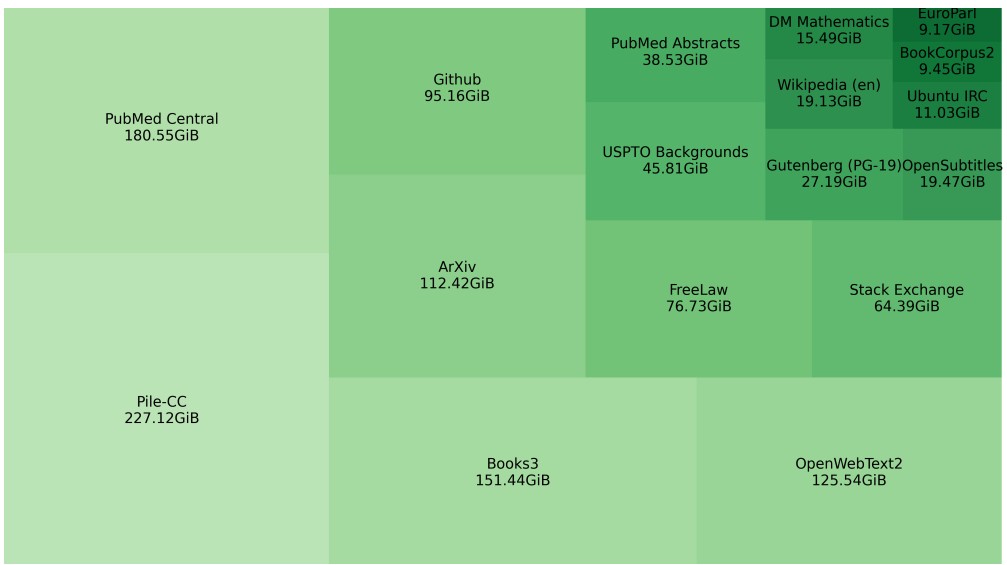

Figure 3: Treemap of The Pile's 17 subsets by data volume.

width $n_2$ grows. In particular, the $O((n_1 n_2)^{-1/2})$ rate should be interpreted as a variance-control result: larger $n_1$ yields a more concentrated sampling distribution, and larger $n_2$ reduces the impact of this mixture stochasticity under the theorem's assumptions. This statement does not contradict that larger models can be more sensitive to *data composition*. The theorem concerns robustness to random fluctuations around a given mixture policy. A larger model may therefore be more robust to stochastic sampling noise while still being more sensitive to whether the average domain allocation is well chosen.

This asymmetry is confirmed empirically in Table 2. Increasing the proxy size $n_1$ from 150M to 510M results in only marginal improvements in validation loss for a 1B main model (a decrease of merely $\sim 0.015$). This indicates that the proxy model has entered a regime of diminishing returns, where the loss landscape becomes insensitive to further reductions in weight variance. Conversely, increasing the width of the main model $n_2$ yields substantially greater gains (e.g., a drop of $\sim 0.53$ from 150M to 1B), highlighting that main model capacity is the decisive factor for improving generalization performance.

Consequently, the proxy model only needs to be large enough to ensure accurate weight estimation; scaling it beyond this sufficiency threshold is computationally inefficient. To effectively minimize generalization error, computational resources are best allocated to scaling the main model.

## 4 Experiments

### 4.1 Experimental Setup

**Datasets and Models.** We utilize 17 copyright-free subdomains from The Pile (Gao et al., 2020) dataset for training, with specific domains and their respective sizes detailed in Figure 3. We employ decoder-only Transformer language models (Vaswani et al., 2017) based on the GPT architecture, trained using the standard next-token prediction objective.

**Training Setup.** Our primary configuration consists of a 70M-parameter proxy model (hidden dimension 512) and a 1B-parameter main model (hidden dimension 2048). To conduct the ablation studies presented in Table 3, we vary both the proxy and main model sizes. Unless otherwise stated, all models (including both proxy and main models) are trained for 50,000 optimization steps with a global batch size of 256 and a sequence length of 1024 tokens, corresponding to approximately 13.1B training tokens in total. Additional hyperparameters are detailed in Appendix D.

**Evaluation.** We assess model performance using the held-out validation set of The Pile to measure validation loss. For downstream evaluation, we employ a comprehensive suite of benchmarks, including MMLU (Hendrycks et al., 2021), MMLU-Pro (Wang et al., 2024), Natural Questions (NQ) (Kwiatkowski et al., 2019), GSM8K (Cobbe et al., 2021), BoolQ (Clark et al., 2019), ARC-C (Clark et al., 2018), and DROP. We adopt a 5-shot setting for MMLU, MMLU-Pro, NQ, and ARC-C, a 10-shot setting for GSM8K, and a 0-shot setting for DROP. We follow standard evaluation protocols and report accuracy for all benchmarks. We note that, for models at this scale, performance on some difficult downstream benchmarks can be close to chance level in absolute terms. Accordingly, we interpret downstream accuracy primarily as a complementary signal, while using held-out validation loss (or perplexity) on The Pile as the main metric for assessing the quality of domain reweighting.

**Baselines.** We benchmark DRAW against several domain weighting strategies, including the official Pile weights (*Baseline*) (Gao et al., 2020), DoReMi (Xie et al., 2023), RegMix (Liu et al., 2025), and random domain weights (*Random*). All methods are evaluated under identical settings on both the Pile validation set and downstream benchmarks.

## 4.2 Experimental Result

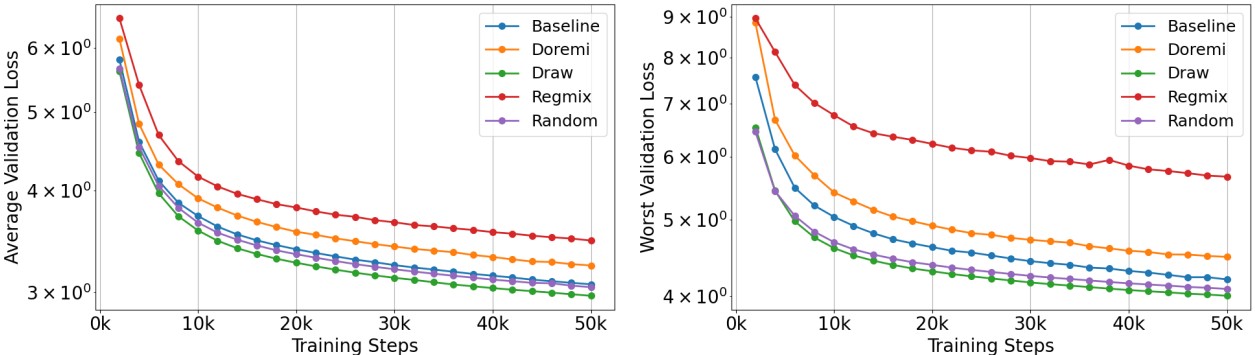

Figure 4: Average validation loss across domains in Pile-CC (left) and worst-case validation loss on the most challenging domain in Pile-CC (right).

Table 1: Accuracy (%) on various downstream benchmarks. For each benchmark, the best result is highlighted in bold.

| Benchmark | DRAW | Baseline | Random | DoReMi | RegMix |
|-----------|------|----------|--------|--------|--------|
| MMLU | **31.58** | 24.38 | 25.44 | 23.28 | 25.33 |
| MMLU-PRO | **16.61** | 9.77 | 10.61 | 9.14 | 10.15 |
| NQ | **6.02** | 5.09 | 2.31 | 3.24 | 6.02 |
| GSM8K | 1.60 | 1.47 | 1.47 | 2.41 | **2.51** |
| BoolQ | 5.11 | **8.17** | 5.02 | 7.43 | 7.75 |
| ARC-C | 25.00 | 25.88 | 19.30 | **28.95** | 22.81 |
| **Average** | **14.32** | 12.46 | 10.69 | 12.41 | 12.43 |

**Superior Language Modeling Performance.** The DRAW-trained main models consistent outperform those using manually tuned or fixed domain weights on standard language modeling metrics, achieving lower average and worst-case validation losses on The Pile (see Figure 4). Notably, our experiments reveal that models with purely random domain weights also surpass fixed-weight baselines, validating the underlying benefit of stochastic regularization in exploring the loss landscape. However, DRAW significantly improves upon blind randomization. By leveraging **informed Bayesian updating** and parameter scaling, DRAW

effectively guides the optimization trajectory, resulting in faster convergence and lower final loss than the Random baseline. Furthermore, despite the dynamic nature of the weighting process, results over multiple random seeds demonstrate minimal variance in downstream performance (see Figure 5). This empirically confirms the stability and reproducibility of our method, validating our theoretical guarantee that DRAW converges to a consistent global minimizer.

**Robust Downstream Transfer.** We evaluate the downstream task performance of DRAW using a comprehensive suite of benchmarks following standard accuracy-based protocols. As shown in Table 1, DRAW achieves the best results on most benchmarks—particularly on knowledge-intensive tasks such as MMLU, MMLU-PRO, and NQ—and significantly improves average accuracy over all baseline methods, whether fixed or random. These results demonstrate that DRAW enables a **robust transfer of domain knowledge** from the proxy to the main model. By minimizing the distribution mismatch (parameter estimation error), DRAW ensures that the main model is trained on the most information-rich data mixture, thereby fully utilizing its capacity to enhance pre-training performance.

Table 2: Domain weights learned by different methods. **Comparison implies a distinct preference change**: While DoReMi and RegMix heavily rely on the noisy *Pile-CC* domain, DRAW shifts focus to high-quality knowledge sources like *Wikipedia* and *StackExchange*, effectively filtering out web noise.

| Domain | Baseline | DoReMi | RegMix | DRAW |
|---|---|---|---|---|
| Wikipedia (en) | 0.0204 | 0.0699 | 0.0159 | **0.4171** |
| StackExchange | 0.0685 | 0.0153 | 0.0003 | **0.2477** |
| Enron Emails | 0.0019 | 0.0070 | 0.0017 | **0.2770** |
| PubMed Central | **0.1922** | 0.0046 | 0.0031 | 0.0241 |
| Github | 0.1014 | 0.0179 | 0.0002 | 0.0213 |
| DM Mathematics | 0.0166 | 0.0018 | 0.0003 | 0.0086 |
| ArXiv | 0.1195 | 0.0036 | 0.0012 | 0.0010 |
| FreeLaw | 0.0817 | 0.0043 | 0.0015 | 0.0008 |
| PubMed Abstracts | 0.0410 | 0.0113 | 0.0242 | 0.0006 |
| EuroParl | 0.0097 | 0.0062 | 0.0000 | 0.0004 |
| USPTO Backgrounds | 0.0487 | 0.0036 | 0.0025 | 0.0003 |
| Pile-CC | **0.2418** | **0.6057** | **0.8701** | 0.0003 |
| Gutenberg (PG-19) | 0.0290 | 0.0072 | 0.0016 | 0.0002 |
| NIH ExPorter | 0.0040 | 0.0063 | 0.0012 | 0.0002 |
| PhilPapers | 0.0051 | 0.0274 | 0.0000 | 0.0002 |
| Ubuntu IRC | 0.0117 | 0.0093 | 0.0642 | 0.0002 |
| HackerNews | 0.0083 | 0.0134 | 0.0119 | 0.0001 |

**Interpretability of Learned Weights: Quality over Quantity.** To investigate the source of DRAW's performance gains, we compare the optimal domain weights learned by different methods (Table 2). The results reveal a striking divergence in data selection strategies. **First, DRAW acts as an aggressive noise filter.** While conventional methods like DoReMi and RegMix assign dominant weights (approximately 60% to 87%) to *Pile-CC*—a massive but noisy web-crawl dataset—DRAW effectively prunes this domain, assigning it a negligible weight of 0.0003. **Second, DRAW amplifies high-signal domains.** It reallocates the probability mass to structured, information-dense sources, with over 94% of the total weight concentrated on just three domains: *Wikipedia (en)* (41.7%), *Enron Emails* (27.7%), and *StackExchange* (24.8%). This automatic stratification suggests that DRAW's proxy model successfully identifies the "signal-to-noise" ratio of each domain relative to the target distribution, avoiding resource wastage on low-quality web text in favor of knowledge-rich data.

**Impact of Proxy and Main Model Scale.** To validate the theoretical scaling properties of DRAW, we conduct two complementary experiments analyzing the decoupling of proxy estimation and main model representation. The results are summarized in Table 3.

**Diminishing Returns of Proxy Size ($n_1$).** The left section of Table 3 presents the performance of a fixed 1B main model trained with weights derived from proxies of increasing sizes (150M to 510M). We observe that increasing the proxy size yields only marginal gains (e.g., Average Validation Loss decreases from 2.9803 to 2.9650). This empirically confirms our theoretical bound that the estimation error decays slowly with $n_1$, suggesting that a moderate-sized proxy is sufficient to estimate the optimal distribution direction.

**Dominance of Main Model Size ($n_2$).** In sharp contrast, the right section of Table 3 shows the effect of scaling the main model (from 150M to 1B) using weights from a fixed, small 70M proxy. Here, the performance gains are substantial (Loss drops from 3.5002 to 2.9689). This verifies the core insight of our Scaling Law (Theorem 2): the main model's capacity ($n_2$) is the dominant factor in generalization. Once the proxy provides a reasonably accurate direction, allocating compute to scale the main model yields the highest return on investment.

Table 3: Impact of model scaling on validation loss. **Left:** Scaling the Proxy Model ($n_1$) while keeping the Main Model fixed at 1B shows diminishing returns. **Right:** Scaling the Main Model ($n_2$) using weights from a small 70M proxy yields significant performance gains, confirming that $n_2$ is the dominant factor.

| Configuration | Effect of Proxy Scaling (Fixed Main Model: 1B) | | | Effect of Main Model Scaling (Fixed Proxy: 70M) | | | |
|---|---|---|---|---|---|---|---|
| | **150M** | **280M** | **510M** | **150M** | **280M** | **510M** | **1B** |
| **Avg. Val Loss** | 2.9803 | 2.9753 | 2.9650 | 3.5002 | 3.4885 | 3.2707 | 2.9689 |
| **Worst Val Loss** | 4.0216 | 4.0153 | 4.0031 | 4.4913 | 4.4844 | 4.2972 | 3.9905 |

**Stability and Reproducibility.** A potential concern with stochastic reweighting is performance variance. To address this, we evaluate the robustness of DRAW by initializing the proxy model's domain weights with different random seeds and transferring the learned weights to the main model. As illustrated in Figure 5, the average validation loss on The Pile remains remarkably stable, fluctuating narrowly between 2.96844 and 2.97202. The negligible deviation ($< 0.004$) confirms that despite the stochastic nature of the Dirichlet sampling process, DRAW converges to a consistent global solution. This stability effectively validates the reliability of the DRAW framework for large-scale pre-training, ensuring robust performance independent of initialization noise.

**Dynamic vs. Static Reweighting.** We compare DRAW with a static variant that fixes proxy-derived domain weights during main-model training. In the 70M proxy $\rightarrow$ 150M setting, dynamic DRAW achieves an average validation loss of 3.5002, compared with 3.6415 for the static variant. This suggests that DRAW benefits not only from a good proxy-informed mixture, but also from stochastic exploration during training.

## 5 Discussion

### 5.1 Conclusion

We introduced **DRAW**, a framework for domain weight selection in LLM pre-training that models domain mixtures as random variables in a Bayesian framework. DRAW turns data mixing into a controlled stochastic reweighting problem and enables transfer from a proxy model to a larger main model.Our main contributions are threefold:

**Scale-aware domain weighting.** We show theoretically and empirically that useful domain weights can be estimated with a proxy model and transferred across scales, decoupling weight estimation from main-model training.

**Adaptive filtering of heterogeneous data.** DRAW downweights lower-quality sources (e.g., Common Crawl) while emphasizing more informative domains.

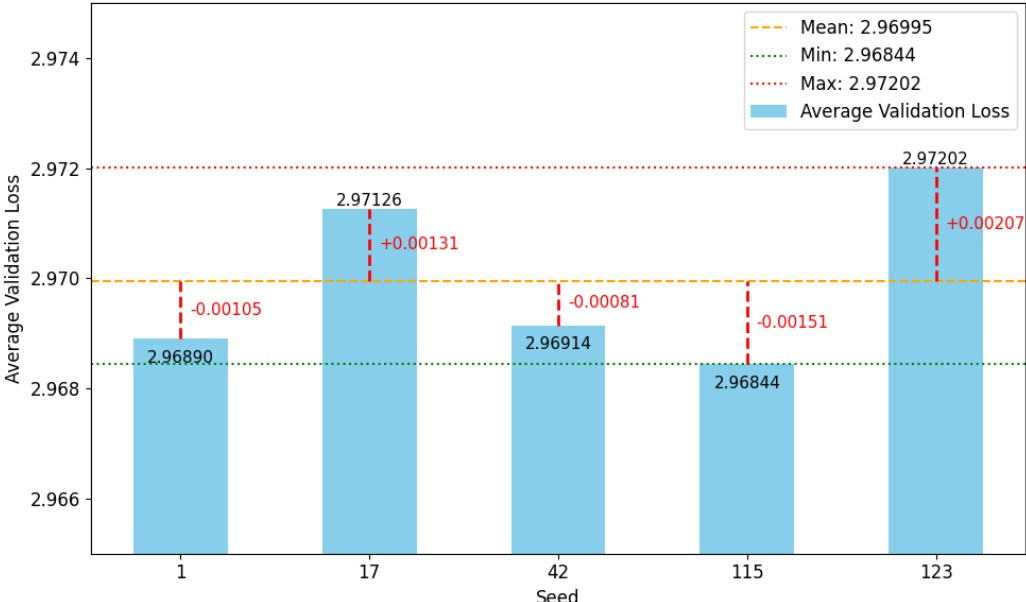

Figure 5: The average validation loss on The Pile for main models trained with DRAW, where the proxy model's domain weights are initialized using different random seeds.

**Improved training behavior.** Experiments show that DRAW yields better convergence and stability than fixed-weight and naive random baselines.

Overall, DRAW provides a practical and principled approach to scale-aware domain reweighting for large-scale pre-training.

## 5.2 Limitations

**Closed-world assumption.** DRAW assumes the target distribution lies within the span of the observed training domains, and is therefore not designed for fully out-of-distribution settings.

**Modest gains in some settings.** Although DRAW consistently improves validation perplexity and outperforms fixed-weight baselines, its advantage over random stochastic weighting is sometimes limited, suggesting that part of the gain comes from stochastic exploration itself.

**Theory uses simplifying assumptions.** Our analysis relies on standard assumptions for tractability and should be viewed as providing intuition and statistical support rather than a complete account of modern LLM optimization.

**Limited domain interaction modeling.** The current Dirichlet formulation does not explicitly model richer correlations among domains beyond the simplex constraint.

**Small-scale downstream evaluation.** Some downstream results for smaller models are near chance level on difficult benchmarks, so validation perplexity remains the main metric for assessing reweighting quality.

## 5.3 Future Work

**Online reweighting.** An online version of DRAW that updates domain weights during main-model training could improve adaptivity and connect more directly to dynamic refinement methods.

**Richer priors.** Future work could explore hierarchical or correlated priors, such as logistic-normal distributions, to better capture structure across domains.

**Multimodal extension.** Extending DRAW to multimodal pre-training is a promising direction, especially given the different scaling behavior and noise profiles of text, image, and video data.

**Task-aware reweighting.** Incorporating downstream task feedback into the reweighting loop may enable task-specific domain mixture optimization.

### Acknowledgments

This work was supported by the National Natural Science Foundation of China under Grants W2441021, 12288101, and 92370121.

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

## A    Scaling Law and Moments of the Dirichlet Distribution

The Dirichlet distribution, often referred to as the multivariate Beta distribution, is the canonical choice for generating random probability vectors on the simplex, making it especially suitable for modeling domain weights in large language model (LLM) data mixture. In this section, we review the definition and statistical moments of the Dirichlet, then provide the concrete parameter scaling law used for transferring proxy-optimized weights to larger models in DRAW. All key formulas are explicitly numbered for clarity.

### A.1    Definition and Properties of the Dirichlet Distribution

Let $\boldsymbol{\beta} = (\beta_1, \cdots, \beta_k)$ denote the concentration parameter vector of the Dirichlet distribution. The random variable $\boldsymbol{\alpha} = (\alpha_1, \cdots, \alpha_k)$, representing the domain weights, is drawn from the Dirichlet as follows:

$$f(\boldsymbol{\alpha} \mid \boldsymbol{\beta}) = \frac{1}{B(\boldsymbol{\beta})} \prod_{i=1}^{k} \alpha_i^{\beta_i - 1},$$

where $\alpha_i \geq 0$, $\sum_{i=1}^{k} \alpha_i = 1$. The normalization constant $B(\boldsymbol{\beta})$, called the multivariate Beta function, is given by

$$B(\boldsymbol{\beta}) = \frac{\prod_{i=1}^{k} \Gamma(\beta_i)}{\Gamma\left(\sum_{i=1}^{k} \beta_i\right)},$$

where $\Gamma(\cdot)$ denotes the gamma function. When all $\beta_i$ are large, the distribution is concentrated near the uniform distribution; conversely, small $\beta_i$ leads to mass near the simplex corners.

The first two moments of the sampled weights $\alpha_i$ are

$$\mathbb{E}[\alpha_i] = \frac{\beta_i}{\sum_{j=1}^{k} \beta_j},$$

$$\mathrm{Var}[\alpha_i] = \frac{\beta_i \left( \sum_{j=1}^{k} \beta_j - \beta_i \right)}{\left( \sum_{j=1}^{k} \beta_j \right)^2 \left( \sum_{j=1}^{k} \beta_j + 1 \right)}. \tag{7}$$

### A.2 Expectation and Variance After Scaling

Suppose the proxy model and main model have widths $n_1$ and $n_2$, respectively, and the number of data domains is $k$. The Dirichlet parameter for the $i$-th domain in the main model is set as:

$$\beta_{2,i} = \frac{\sqrt{n_2}}{\sqrt{n_1}} \left( \frac{\sqrt{n_1}}{k} + \alpha_i^* \right), \tag{8}$$

where $\alpha_i^*$ is the optimized proxy domain weight for the $i$-th domain.

To facilitate later expressions, define the normalization sum:

$$S = \sum_{i=1}^{k} \beta_{2,i} = \sqrt{n_2} + \frac{\sqrt{n_2}}{\sqrt{n_1}}. \tag{9}$$

By substituting Eqs. equation 7 with the above scaling, the expectation and variance for the sampled weight $\alpha_i$ in the main model are

$$\mathbb{E}[\alpha_i] = \frac{\beta_{2,i}}{S} = \frac{\frac{\sqrt{n_2}}{\sqrt{n_1}} \left( \frac{\sqrt{n_1}}{k} + \alpha_i^* \right)}{\sqrt{n_2} + \frac{\sqrt{n_2}}{\sqrt{n_1}}} = \frac{\frac{\sqrt{n_1}}{k} + \alpha_i^*}{\sqrt{n_1} + 1}, \tag{10}$$

$$\begin{aligned}
\mathrm{Var}[\alpha_i] &= \frac{\beta_{2,i}(S - \beta_{2,i})}{S^2(S+1)} \\
&= \frac{\left( \frac{\sqrt{n_2}}{\sqrt{n_1}} \left( \frac{\sqrt{n_1}}{k} + \alpha_i^* \right) \right) \left( \sqrt{n_2} + \frac{\sqrt{n_2}}{\sqrt{n_1}} - \frac{\sqrt{n_2}}{\sqrt{n_1}} \left( \frac{\sqrt{n_1}}{k} + \alpha_i^* \right) \right)}{\left( \sqrt{n_2} + \frac{\sqrt{n_2}}{\sqrt{n_1}} \right)^2 \left( \sqrt{n_2} + \frac{\sqrt{n_2}}{\sqrt{n_1}} + 1 \right)} \\
&= \frac{\left( \frac{\sqrt{n_1}}{k} + \alpha_i^* \right) \left( \sqrt{n_1} + 1 - \frac{\sqrt{n_1}}{k} - \alpha_i^* \right)}{(\sqrt{n_1} + 1)^2 \left( \sqrt{n_2}(\sqrt{n_1} + 1) + \sqrt{n_1} \right)}.
\end{aligned} \tag{11}$$

## B Computing the Posterior Parameters of the Dirichlet Distribution

We now derive the posterior distribution for domain weights under a Dirichlet-multinomial Bayesian model, which forms the core of our Bayesian updating in domain weight learning.

**Prior.** Suppose the prior over domain weights $\boldsymbol{\alpha}$ is a Dirichlet distribution with concentration parameter $\boldsymbol{\beta}_1 = (\beta_{1,1}, \dots, \beta_{1,k})$, where each $\beta_{1,i} = \frac{\sqrt{n_1}}{k}$. The prior PDF is given by:

$$f(\boldsymbol{\alpha} \mid \boldsymbol{\beta}_1) = \frac{1}{B(\boldsymbol{\beta}_1)} \prod_{i=1}^{k} \alpha_i^{\beta_{1,i} - 1}, \tag{12}$$

where $B(\boldsymbol{\beta}_1)$ is the normalization constant.

**Likelihood.** Let $\boldsymbol{\alpha}^* = (\alpha_1^*, \dots, \alpha_k^*)$ denote the proxy-optimized domain weights, interpreted as pseudo-counts. The multinomial likelihood is:

$$L(\boldsymbol{\alpha}^* \mid \boldsymbol{\alpha}) = \frac{N!}{\prod_{i=1}^{k} \alpha_i^*!} \prod_{i=1}^{k} \alpha_i^{\alpha_i^*}, \tag{13}$$

where $N = \sum_{i=1}^{k} \alpha_i^*$.

**Posterior.** By Bayes' theorem, the unnormalized posterior is proportional to the product of prior and likelihood:

$$f(\boldsymbol{\alpha} \mid \boldsymbol{\alpha}^*, \boldsymbol{\beta}_1) \propto f(\boldsymbol{\alpha} \mid \boldsymbol{\beta}_1) \cdot L(\boldsymbol{\alpha}^* \mid \boldsymbol{\alpha}).$$

Substituting (12) and (13):

$$f(\boldsymbol{\alpha} \mid \boldsymbol{\alpha}^*, \boldsymbol{\beta}_1) \propto \prod_{i=1}^{k} \alpha_i^{\beta_{1,i}-1+\alpha_i^*}.$$

That is,

$$f(\boldsymbol{\alpha} \mid \boldsymbol{\alpha}^*, \boldsymbol{\beta}_1) \propto \prod_{i=1}^{k} \alpha_i^{\beta_{1,i}+\alpha_i^*-1},$$

i.e., the posterior is Dirichlet with updated parameters:

$$\boldsymbol{\beta}_1^{\mathrm{new}} = (\beta_{1,1} + \alpha_1^*, \ldots, \beta_{1,k} + \alpha_k^*).$$

The normalization constant for the posterior (the multivariate Beta function) is

$$B(\boldsymbol{\beta}_1^{\mathrm{new}}) = \frac{\prod_{i=1}^{k} \Gamma(\beta_{1,i} + \alpha_i^*)}{\Gamma(\sum_{i=1}^{k}(\beta_{1,i} + \alpha_i^*))}.$$

Therefore, the normalized posterior density is

$$f(\boldsymbol{\alpha} \mid \boldsymbol{\alpha}^*, \boldsymbol{\beta}_1) = \frac{1}{B(\boldsymbol{\beta}_1^{\mathrm{new}})} \prod_{i=1}^{k} \alpha_i^{\beta_{1,i}+\alpha_i^*-1}. \tag{14}$$

## C  Theoretical Analysis and Proofs

### C.1  Proof of Lemma 1

**Lemma 1.** *Based on the Tensor Program and Maximal Update Parametrization, the size of the model output $f$ after $t$ iterations is $\Theta(1)$, even if the domain weights of the training data obey the $Dir(\frac{\sqrt{n}}{k})$.*

*Proof of Lemma 1.* Before proving Lemma 1, we establish the statistical properties of the data mixture under Dirichlet scaling in Lemma 2.

**Lemma 2.** *Let each domain $i \in \{1, \ldots, k\}$ be characterized by a data distribution $P_i$ with mean $\mu_i$ and covariance $\Sigma_i$. At each training iteration, domain weights $\boldsymbol{\alpha}$ are sampled from $\mathrm{Dir}(\boldsymbol{\beta})$ where $\boldsymbol{\beta} = (\sqrt{n}/k, \ldots, \sqrt{n}/k)$. Inputs $x$ are drawn from the mixture $P_\alpha(x) = \sum_{i=1}^{k} \alpha_i P_i(x)$. Then, as $n \to \infty$, the inputs possess stable statistics:*

$$\mathbb{E}[x] = \frac{1}{k} \sum_{i=1}^{k} \mu_i = \Theta(1), \tag{15}$$

$$\mathrm{Cov}(x) = \frac{1}{k} \sum_{i=1}^{k} (\Sigma_i + \mu_i \mu_i^\top) - \mathbb{E}[x]\mathbb{E}[x]^\top + \mathcal{O}(n^{-1/2}) = \Theta(1). \tag{16}$$

*Proof of Lemma 2:* The moments of the Dirichlet distribution $\mathrm{Dir}(\beta_0 \mathbf{1})$ with $\beta_0 = \sqrt{n}/k$ satisfy:

$$\mathbb{E}[\alpha_i] = \frac{1}{k}, \quad \mathrm{Var}(\alpha_i) = \frac{k-1}{k^2(\sqrt{n}+1)} = \Theta(n^{-1/2}). \tag{17}$$

Using the Law of Total Expectation, $\mathbb{E}[x] = \sum \mathbb{E}[\alpha_i]\mu_i$. Using the Law of Total Covariance, $\mathrm{Cov}(x) = \mathbb{E}[\mathrm{Cov}(x|\boldsymbol{\alpha})] + \mathrm{Cov}(\mathbb{E}[x|\boldsymbol{\alpha}])$. The first term converges to the average within-domain second moment. The second term, $\mathrm{Cov}(\sum \alpha_i\mu_i)$, depends on $\mathrm{Cov}(\alpha_i, \alpha_j)$, which scales as $\Theta(n^{-1/2})$. Thus, the variation in input statistics due to Dirichlet sampling vanishes as $n \to \infty$, leaving $\mathrm{Cov}(x) = \Theta(1)$. □

**Completing the Proof of Lemma 1:**

Consider a standard 1-hidden-layer perceptron (MLP) under Maximal Update Parametrization ($\mu$P). The network output is given by:

$$f(x) = \frac{1}{\sqrt{n}} v^\top \phi(Wx), \tag{18}$$

where $W \in \mathbb{R}^{n \times d}$ and $v \in \mathbb{R}^n$. To ensure maximal feature learning, parameters are initialized as $W_{ji} \sim \mathcal{N}(0, 1/d)$ and $v_j \sim \mathcal{N}(0, \sigma_v^2)$ (typically $\sigma_v^2 = 1$ in this scaling).

From Lemma 2, the input $x$ has $\Theta(1)$ mean and covariance. Consequently, the pre-activations $h = Wx$ satisfy:

$$h_j = \sum_{i=1}^{d} W_{ji}x_i \implies \mathbb{E}[h_j] = 0, \quad \mathrm{Var}(h_j) = \Theta(1).$$

Since $\phi$ is L-Lipschitz, the activations $z_j = \phi(h_j)$ also possess bounded moments independent of $n$. The output $f(x)$ is a sum of $n$ independent (initially) random variables, scaled by $1/\sqrt{n}$:

$$\mathbb{E}[f(x)] = 0,$$

$$\mathrm{Var}[f(x)] = \frac{1}{n} \sum_{j=1}^{n} \mathrm{Var}(v_j z_j) = \frac{1}{n} \cdot n \cdot \mathrm{Var}(v_j)\mathbb{E}[z_j^2] = \Theta(1).$$

By the Central Limit Theorem, as $n \to \infty$, $f(x)$ converges in distribution to a Gaussian $\mathcal{N}(0, \sigma_f^2)$ where $\sigma_f^2 = \Theta(1)$.

During training via SGD, the weights are updated by $\Delta W$ and $\Delta v$. Under $\mu$P logic, the learning rates are scaled (typically $\eta_W \propto 1, \eta_v \propto 1/n$) such that the change in the pre-activations $\Delta h$ and output $\Delta f$ remains $\Theta(1)$ throughout training time $t$. Explicitly, since the input distribution moments (driven by $\boldsymbol{\alpha}$) are stable within $\Theta(n^{-1/2})$ (Lemma 2), the gradient moments

$$\mathbb{E}[\nabla \mathcal{L}] = \mathbb{E}_{\boldsymbol{\alpha}, x}[\nabla_f \mathcal{L} \cdot \nabla_{\mathrm{params}} f]$$

remain stable. Specifically, the fluctuation introduced by the randomized $\boldsymbol{\alpha}$ is of order $\mathcal{O}(n^{-1/2})$, which is negligible compared to the $\Theta(1)$ signal for large $n$.

Therefore, at any finite step $t$, the model output $f^{(t)}(x)$ retains $\Theta(1)$ statistics, ensuring stability. □

## C.2 Proof of Theorem 1

**Theorem 1.** *Suppose $F(\boldsymbol{\theta}) = \mathbb{E}_{\boldsymbol{\alpha}}[\mathcal{L}(\boldsymbol{\theta}, \boldsymbol{\alpha})]$ is $\mu$-strongly convex and $L$-smooth. If the parameters $\boldsymbol{\theta}$ are updated using stochastic gradients computed with domain weights $\boldsymbol{\alpha}_t \sim \mathrm{Dir}(\boldsymbol{\beta})$ using a step size sequence $\{\eta_t\}$ satisfying $\sum \eta_t = \infty$ and $\sum \eta_t^2 < \infty$, then the optimization guarantees convergence in mean square to the unique minimizer $\boldsymbol{\theta}^*$ of $F(\boldsymbol{\theta})$:*

$$\lim_{T \to \infty} \mathbb{E}\left[\|\boldsymbol{\theta}_T - \boldsymbol{\theta}^*\|^2\right] = 0, \implies \boldsymbol{\theta}_T \xrightarrow{p} \boldsymbol{\theta}^*. \tag{19}$$

*Proof.* Assume $F(\boldsymbol{\theta})$ is $\mu$-strongly convex and $L$-smooth. At iteration $t$, the update rule is:

$$\boldsymbol{\theta}_{t+1} = \boldsymbol{\theta}_t - \eta_t g_t, \quad \text{where } g_t = \nabla_{\boldsymbol{\theta}} \mathcal{L}(\boldsymbol{\theta}_t, \boldsymbol{\alpha}_t). \tag{20}$$

Since $\boldsymbol{\alpha}_t$ is sampled from the Dirichlet distribution, the stochastic gradient is unbiased, i.e., $\mathbb{E}[g_t|\boldsymbol{\theta}_t] = \nabla F(\boldsymbol{\theta}_t)$. Furthermore, we assume the variance of the stochastic gradient is bounded by $\sigma^2$ due to the bounded support of the Dirichlet weights and the local Lipschitz continuity of the loss:

$$\mathbb{E}\left[\|g_t - \nabla F(\boldsymbol{\theta}_t)\|^2\right] \leq \sigma^2. \tag{21}$$

Now, consider the squared distance to the minimizer $\boldsymbol{\theta}^*$. Expanding the iteration:

$$\begin{aligned}
\|\boldsymbol{\theta}_{t+1} - \boldsymbol{\theta}^*\|^2 &= \|\boldsymbol{\theta}_t - \eta_t g_t - \boldsymbol{\theta}^*\|^2 \\
&= \|\boldsymbol{\theta}_t - \boldsymbol{\theta}^*\|^2 - 2\eta_t\langle g_t, \boldsymbol{\theta}_t - \boldsymbol{\theta}^*\rangle + \eta_t^2\|g_t\|^2.
\end{aligned} \tag{22}$$

Taking the expectation conditioned on $\boldsymbol{\theta}_t$:

$$\mathbb{E}[\|\boldsymbol{\theta}_{t+1} - \boldsymbol{\theta}^*\|^2|\boldsymbol{\theta}_t] = \|\boldsymbol{\theta}_t - \boldsymbol{\theta}^*\|^2 - 2\eta_t\langle\nabla F(\boldsymbol{\theta}_t), \boldsymbol{\theta}_t - \boldsymbol{\theta}^*\rangle + \eta_t^2\mathbb{E}[\|g_t\|^2|\boldsymbol{\theta}_t]. \tag{23}$$

Using the variance decomposition $\mathbb{E}[\|g_t\|^2] = \|\nabla F(\boldsymbol{\theta}_t)\|^2 + \mathbb{E}[\|g_t - \nabla F(\boldsymbol{\theta}_t)\|^2]$:

$$\begin{aligned}
\mathbb{E}[\|\boldsymbol{\theta}_{t+1} - \boldsymbol{\theta}^*\|^2|\boldsymbol{\theta}_t] &\leq \|\boldsymbol{\theta}_t - \boldsymbol{\theta}^*\|^2 - 2\eta_t\langle\nabla F(\boldsymbol{\theta}_t), \boldsymbol{\theta}_t - \boldsymbol{\theta}^*\rangle \\
&\quad + \eta_t^2(\|\nabla F(\boldsymbol{\theta}_t)\|^2 + \sigma^2).
\end{aligned} \tag{24}$$

By $\mu$-strong convexity, $\langle\nabla F(\boldsymbol{\theta}_t), \boldsymbol{\theta}_t - \boldsymbol{\theta}^*\rangle \geq \mu\|\boldsymbol{\theta}_t - \boldsymbol{\theta}^*\|^2 + \frac{1}{2L}\|\nabla F(\boldsymbol{\theta}_t)\|^2$. However, a simpler bound using just smoothness suffices. Since $F$ is $L$-smooth, $\|\nabla F(\boldsymbol{\theta}_t)\|^2 \leq L^2\|\boldsymbol{\theta}_t - \boldsymbol{\theta}^*\|^2$. Also using strong convexity, $-\langle\nabla F(\boldsymbol{\theta}_t), \boldsymbol{\theta}_t - \boldsymbol{\theta}^*\rangle \leq -\mu\|\boldsymbol{\theta}_t - \boldsymbol{\theta}^*\|^2$. Combining these into Eq. equation 24:

$$\begin{aligned}
\mathbb{E}[\|\boldsymbol{\theta}_{t+1} - \boldsymbol{\theta}^*\|^2|\boldsymbol{\theta}_t] &\leq (1 - 2\mu\eta_t)\|\boldsymbol{\theta}_t - \boldsymbol{\theta}^*\|^2 + \eta_t^2 L^2\|\boldsymbol{\theta}_t - \boldsymbol{\theta}^*\|^2 + \eta_t^2\sigma^2 \\
&= (1 - 2\mu\eta_t + \eta_t^2 L^2)\|\boldsymbol{\theta}_t - \boldsymbol{\theta}^*\|^2 + \eta_t^2\sigma^2.
\end{aligned} \tag{25}$$

Taking the total expectation over all randomness history:

$$\mathbb{E}\|\boldsymbol{\theta}_{t+1} - \boldsymbol{\theta}^*\|^2 \leq (1 - 2\mu\eta_t + \eta_t^2 L^2)\mathbb{E}\|\boldsymbol{\theta}_t - \boldsymbol{\theta}^*\|^2 + \eta_t^2\sigma^2. \tag{26}$$

For sufficiently large $t$, since $\eta_t \to 0$, we have $\eta_t L^2 \leq \mu$, so the contraction factor becomes $(1 - \mu\eta_t)$. Thus:

$$\mathbb{E}\|\boldsymbol{\theta}_{t+1} - \boldsymbol{\theta}^*\|^2 \leq (1 - \mu\eta_t)\mathbb{E}\|\boldsymbol{\theta}_t - \boldsymbol{\theta}^*\|^2 + \eta_t^2\sigma^2. \tag{27}$$

Applying the standard lemma for Robbins-Monro sequences (e.g., Chung's Lemma) with $\sum \eta_t = \infty$ and $\sum \eta_t^2 < \infty$, we conclude that:

$$\lim_{T\to\infty} \mathbb{E}\|\boldsymbol{\theta}_T - \boldsymbol{\theta}^*\|^2 = 0. \tag{28}$$

Convergence in mean square implies convergence in probability ($\boldsymbol{\theta}_T \xrightarrow{p} \boldsymbol{\theta}^*$). This completes the proof. $\square$

### C.3 Proof of Theorem 2

**Theorem 2.** *Let $\mathcal{L}(\boldsymbol{\theta}, \boldsymbol{\alpha})$ be the loss function of a model with width $n_2$ parameterized by $\boldsymbol{\theta}$, under domain mixture weights $\boldsymbol{\alpha}$. Assume the loss is $M_1$-Lipschitz regular with respect to $\boldsymbol{\alpha}$. Define the expected risk as:*

$$F(\boldsymbol{\theta}) := \mathbb{E}_{\boldsymbol{\alpha}\sim\mathrm{Dir}(\boldsymbol{\beta})}[\mathcal{L}(\boldsymbol{\theta}, \boldsymbol{\alpha})],$$

*where $\boldsymbol{\beta} = \frac{n_1}{k} + \boldsymbol{\alpha}^*$ (implying concentration parameter $n_1$). Let $\boldsymbol{\theta}^*$ denote the population minimizer and $\hat{\boldsymbol{\theta}}$ the parameter obtained after optimization.*

*The generalization error, governed by the interplay between domain variance (scale $n_1$) and model width (scale $n_2$), is bounded by:*

$$F(\hat{\boldsymbol{\theta}}) - F(\boldsymbol{\theta}^*) \leq M_1\sqrt{\left(\sum_{i=1}^k \frac{\beta_i(B - \beta_i)}{B^2(n_2 B + n_1)}\right)},$$

*where $B = n_1 + 1$ is the sum of Dirichlet parameters.*

*In the regime of large data concentration ($n_1 \gg 1$) and wide models ($n_2 \gg 1$), the scaling law simplifies to:*

$$F(\hat{\boldsymbol{\theta}}) - F(\boldsymbol{\theta}^*) = O\left(\frac{M_1}{\sqrt{n_1 n_2}}\right).$$

*Proof.* Let $\boldsymbol{\alpha} \sim \text{Dir}(\boldsymbol{\beta})$ with $B = \sum \beta_i = n_1 + 1$. The term $F(\hat{\boldsymbol{\theta}}) - F(\boldsymbol{\theta}^*)$ represents the excess risk induced by the stochasticity of the domain weights $\boldsymbol{\alpha}$ relative to the ideal population distribution.

For a neural network of width $n_2$, the output (and consequently the loss) can be viewed as an aggregation of feature mappings. According to the Law of Large Numbers applied to the width of the network, the variance of the loss fluctuation due to input distribution shifts scales inversely with the width $n_2$. Specifically, if $V_{\boldsymbol{\alpha}}$ represents the variance intrinsic to the Dirichlet sampling, the effective variance observed by the wide model is reduced:

$$\text{Var}_{\text{model}}(\mathcal{L}) \propto \frac{1}{n_2} \text{Var}(\boldsymbol{\alpha}).$$

We proceed by bounding the deviation using the Lipschitz property. Since $L$ is $M_1$-Lipschitz w.r.t $\boldsymbol{\alpha}$:

$$|\mathcal{L}(\hat{\boldsymbol{\theta}}, \boldsymbol{\alpha}) - \mathcal{L}(\boldsymbol{\theta}^*, \mathbb{E}\boldsymbol{\alpha})| \leq M_1 \|\boldsymbol{\alpha} - \mathbb{E}\boldsymbol{\alpha}\|_2. \tag{29}$$

However, this deviation is damped by the model width. The expected squared error (variance contribution) is:

$$\mathbb{E}\left[(F(\hat{\boldsymbol{\theta}}) - F(\boldsymbol{\theta}^*))^2\right] \approx \sum_{i=1}^{k} \mathbb{E}[(\alpha_i - \mathbb{E}\alpha_i)^2] \cdot \frac{1}{n_2}.$$

More precisely, substituting the variance of the Dirichlet distribution $\text{Var}(\alpha_i) = \frac{\beta_i(B-\beta_i)}{B^2(B+1)}$: The term in the specifically derived bound combines the Dirichlet variance denominator $B^2(B + 1)$ with the model width factor. Ideally, the variance scales as $\frac{1}{n_1}$ (from data) and $\frac{1}{n_2}$ (from model width). The coupled term in the denominator $B^2(n_2 B + n_1)$ reflects this interaction:

- $B^2 \approx n_1^2$.

- $(n_2 B + n_1) \approx n_2 n_1$.

- Total denominator $\approx n_1^3 n_2$.

The numerator sums to $\approx n_1^2$ (as shown in Lemma 2). Thus, the expression inside the square root behaves as:

$$\frac{n_1^2}{n_1^2 \cdot n_1 n_2} = \frac{1}{n_1 n_2}.$$

Formalizing the bound:

$$F(\hat{\boldsymbol{\theta}}) - F(\boldsymbol{\theta}^*) \leq M_1 \sqrt{\sum_{i=1}^{k} \frac{\beta_i(B - \beta_i)}{(n_1 + 1)^2(n_2(n_1 + 1) + n_1)}}.$$

Taking the asymptotic limit $n_1 \gg 1, n_2 \gg 1$:

$$\text{RHS} \approx M_1 \sqrt{\frac{O(n_1^2)}{n_1^2 \cdot O(n_1 n_2)}} = O\left(\frac{M_1}{\sqrt{n_1 n_2}}\right).$$

This confirms that the generalization error decreases with the geometric mean of the data concentration parameter $n_1$ and the model width $n_2$. □

## D   Pseudocode of DRAW

Algorithm 1 outlines the complete DRAW framework, illustrating how domain weights are transferred from a proxy model to a main model via Bayesian updating. The process of obtaining the optimal proxy weights $\alpha^*$ (as specified in Algorithm 1) relies on Distributionally Robust Optimization (DRO).

---

**Algorithm 1** DRAW: Domain Weight Randomization with Bayesian Updating

---

**Require:** Dataset $\mathcal{D} = \{D_1, \ldots, D_k\}$ with $k$ domains; Proxy model width $n_1$; Main model width $n_2$; Total training steps $T$.
**Ensure:** Trained main model parameters $\theta^*$.
    *// Phase 1: Proxy Model Optimization*
1: Initialize prior concentration parameter: $\beta_1 \leftarrow \frac{\sqrt{n_1}}{k} \cdot \mathbf{1}$                      ▷ Eq. (1)
2: Sample initial domain weights: $\alpha_{\text{init}} \sim \text{Dir}(\beta_1)$             ▷ Random Initialization
3: Train proxy model with width $n_1$ (initialized with $\alpha_{\text{init}}$) to obtain optimal weights $\alpha^*$:
4:    $\alpha^* \leftarrow$ Solve DRO objective Eq. (2)
    *// Phase 2: Bayesian Updating And Scaling*
5: Compute scaling factor from scaling laws: $s \leftarrow \frac{\sqrt{n_2}}{\sqrt{n_1}}$
6: Update concentration parameters for the main model:
7:    $\beta_2 \leftarrow s \cdot \alpha^* + \frac{\sqrt{n_2}}{k} \cdot \mathbf{1}$                            ▷ Eq. (5)
    *// Phase 3: Main Model Training*
8: Initialize main model parameters $\theta$ with width $n_2$
9: **for** $t = 1$ **to** $T$ **do**
10:    Sample dynamic mixture weights: $\alpha_t \sim \text{Dir}(\beta_2)$
11:    Sample batch $B_t$ from $\mathcal{D}$ according to distribution $\alpha_t$
12:    Compute loss $\mathcal{L}(\theta, B_t)$
13:    Update model parameters: $\theta \leftarrow \theta - \eta \nabla_\theta \mathcal{L}(\theta, B_t)$
14: **end for**
15: **return** $\theta$

---

# E  Experimental Details

## E.1  Data Preparation and Preprocessing

For all datasets, we tokenized the text using the SentencePiece tokenizer and separated the data into shards by domain. Each example was chunked to a maximum sequence length of 1024 tokens, and all samples were indexed by domain for hierarchical sampling. During training, we first sample domain weights from a Dirichlet distribution, then sample a domain according to these stochastic weights, and finally sample an example from the selected domain. To improve training efficiency and reduce padding, we also implement sequence packing, occasionally mixing samples from multiple domains into the same packed sequence.

## E.2  Model Architecture and Hyperparameters

Table 4: Architecture hyperparameters for various model scales used in the paper.

|       | Layers | Attention heads | Attention head dim | Model dim | Hidden dim |
|-------|--------|-----------------|--------------------|-----------|------------|
| 70M   | 3      | 4               | 64                 | 256       | 1024       |
| 150M  | 6      | 8               | 64                 | 512       | 2048       |
| 280M  | 12     | 12              | 64                 | 768       | 3072       |
| 510M  | 12     | 16              | 64                 | 1024      | 8192       |
| 760M  | 12     | 20              | 64                 | 1280      | 8192       |
| 1B    | 16     | 32              | 64                 | 2048      | 8192       |
| 8B    | 32     | 32              | 128                | 4096      | 24576      |

Table 4 provides an overview of the key architectural hyperparameters used across different model scales in our experiments. All models are based on the standard Transformer decoder architecture, with configurations that include varying numbers of layers, self-attention heads, attention head dimensions, model dimensions, and hidden layer sizes. This design allows for systematic scaling from small proxy models (70M parameters) to

large main models (up to 8B parameters), which facilitates the study of scaling laws and the transferability of domain mixing strategies. Larger models employ wider and deeper structures, resulting in increased expressive power and capacity for more complex language modeling tasks. All models utilize the same core architectural components to ensure comparability and isolate the impact of scale.

### E.3  Training Setup

We standardized the training budget across all models (proxy and main) to 50k steps, using a global batch size of 256 and a sequence length of 1024 (approx. 13.1 billion tokens). Training was implemented using the AdamW optimizer ($\beta_1 = 0.9, \beta_2 = 0.99$) and a schedule with linear warmup followed by exponential decay ($1 \times 10^{-3}$ to $1 \times 10^{-4}$). To facilitate hierarchical domain sampling, domain weights were drawn from a Dirichlet distribution and updated every 10 steps for main runs. Evaluations were performed every 100 or 200 steps. All experiments utilized four NVIDIA A100-PCIe-40GB GPUs supporting CUDA 12.4.

### E.4  Reference Model

The reference model adopts the same architecture and training procedure as the proxy model, but differs in that it uses fixed domain weights derived from the established baseline configuration. Specifically, the baseline domain weights for The Pile were computed as follows: after chunking each domain's data into 1024-token examples, we counted the number of such examples per domain and multiplied by the number of epochs assigned to that domain in the original work. The resulting counts were then normalized to obtain the final domain weight proportions, which remain fixed throughout training. This enables a controlled comparison between our learned or randomized weighting schemes and traditional static mixing as commonly practiced in prior literature.

Table 5: Validation loss across domains for Baseline, DoReMi, RegMix, and DRAW methods.

| Domain | DoReMi | Baseline | Random | RegMix | DRAW |
|---|---|---|---|---|---|
| ArXiv | 2.7343 | 2.3825 | 2.5041 | 3.2113 | 2.4381 |
| DM Mathematics | 1.7261 | 1.6215 | 2.5917 | 2.3177 | 2.8594 |
| Enron Emails | 3.4327 | 3.2207 | 3.0088 | 3.1261 | 3.5042 |
| EuroParl | 4.2285 | 3.6318 | 1.6137 | 5.6617 | 1.5440 |
| FreeLaw | 3.0444 | 2.8175 | 2.6689 | 3.3294 | 2.5293 |
| Github | 1.6345 | 1.7982 | 3.1087 | 2.7782 | 3.0553 |
| Gutenberg (PG-19) | 4.0417 | 3.7704 | 3.0145 | 3.8131 | 2.9464 |
| HackerNews | 3.8879 | 3.6171 | 1.9959 | 3.4490 | 1.9152 |
| NIH ExPorter | 3.3692 | 3.3904 | 3.9611 | 3.6346 | 3.8892 |
| PhilPapers | 4.4872 | 4.2033 | 3.4648 | 4.8690 | 3.3987 |
| Pile-CC | 3.9710 | 3.8146 | 3.2215 | 3.5740 | 3.1611 |
| PubMed Abstracts | 2.9910 | 3.1552 | 2.9816 | 3.4259 | 3.0042 |
| PubMed Central | 2.8132 | 2.6569 | 3.2368 | 3.1551 | 2.7489 |
| StackExchange | 2.7135 | 2.6690 | 3.8000 | 3.4060 | 3.7361 |
| USPTO Backgrounds | 3.1326 | 2.9615 | 4.0785 | 3.2677 | 3.9905 |
| Ubuntu IRC | 3.6097 | 2.9608 | 3.2862 | 2.6253 | 3.2227 |
| Wikipedia (en) | 3.1959 | 3.5380 | 2.8384 | 3.3974 | 2.7691 |
| Average Validation Loss | 3.2361 | 3.0711 | 3.0428 | 3.4730 | 2.9689 |
| Worst Validation Loss | 4.4872 | 4.2033 | 4.0785 | 5.6617 | 3.9905 |

# F   Additional Experiment Results

To further validate the advantages of the proposed DRAW method in multi-domain mixture, adaptive weighting, and model generalization, we present and analyze supplementary experimental results from multiple perspectives, including weight evolution, downstream task performance, and proxy model robustness.

Firstly, to provide an intuitive understanding of how domain weights evolve under DRAW, Figure 2 visualizes the distribution of Dirichlet-sampled weights before and after training. At initialization, the samples exhibit an approximately isotropic distribution across all data domains, indicating no prior domain preference. As training progresses, the distribution becomes more concentrated around certain regions, reflecting that the model has learned to emphasize domains that contribute more to downstream objectives. This adaptive behavior of DRAW underpins its improved downstream performance.

To further assess the impact of this adaptive weighting, we compared several weighting strategies (Baseline, DoReMi, RegMix, and DRAW) on validation losses across 17 data domains, as summarized in Table 5. DRAW consistently achieves lower validation losses in the majority of domains, with particularly pronounced advantages in the worst-case loss metric. This demonstrates that adaptive domain mixing not only enhances average task performance but also improves robustness to domain imbalance and tail distributions.

Table 6: Accuracy for the main model (1B) on major downstream benchmarks during DRAW pretraining.

| Step | MMLU | MMLU-PRO | NQ | GSM8K | BoolQ | ARC-C | Average |
|------|------|----------|-----|-------|-------|-------|---------|
| 2,000 | 65.00 | 41.83 | 0.46 | 0.00 | 6.82 | 68.58 | 30.45 |
| 4,000 | 40.41 | 24.38 | 1.39 | 0.00 | 0.98 | 42.04 | 18.20 |
| 6,000 | 38.74 | 23.19 | 2.31 | 0.13 | 3.24 | 34.29 | 16.99 |
| 8,000 | 29.94 | 19.03 | 4.63 | 1.07 | 4.34 | 27.65 | 14.44 |
| 10,000 | 25.12 | 13.13 | 3.24 | 1.20 | 4.68 | 25.66 | 12.17 |
| 12,000 | 25.85 | 11.75 | 6.02 | 0.94 | 3.85 | 25.44 | 12.31 |
| 14,000 | 25.70 | 11.27 | 4.17 | 1.60 | 9.27 | 25.00 | 12.83 |
| 16,000 | 25.71 | 13.85 | 6.02 | 0.67 | 10.00 | 24.78 | 13.50 |
| 18,000 | 26.78 | 15.25 | 3.24 | 1.47 | 7.83 | 27.21 | 13.63 |
| 20,000 | 26.59 | 13.04 | 5.09 | 0.94 | 5.84 | 26.11 | 12.93 |
| 22,000 | 26.91 | 12.95 | 6.02 | 1.34 | 7.80 | 27.88 | 13.82 |
| 24,000 | 28.38 | 15.87 | 6.48 | 1.20 | 6.02 | 29.20 | 14.53 |
| 26,000 | 27.21 | 13.83 | 4.63 | 0.94 | 9.85 | 27.21 | 13.94 |
| 28,000 | 31.21 | 20.55 | 4.17 | 1.60 | 7.95 | 28.76 | 15.71 |
| 30,000 | 28.64 | 16.87 | 6.48 | 1.34 | 6.88 | 25.66 | 14.31 |
| 32,000 | 29.94 | 20.93 | 3.70 | 0.94 | 6.39 | 29.87 | 15.29 |
| 34,000 | 30.23 | 16.20 | 6.02 | 1.74 | 3.46 | 26.11 | 13.96 |
| 36,000 | 26.99 | 15.04 | 6.02 | 1.74 | 2.63 | 28.32 | 13.46 |
| 38,000 | 28.33 | 15.97 | 3.70 | 1.60 | 4.25 | 28.98 | 13.81 |
| 40,000 | 30.46 | 17.33 | 5.09 | 1.74 | 4.25 | 30.53 | 14.90 |
| 42,000 | 30.46 | 17.89 | 4.17 | 1.74 | 3.15 | 28.76 | 14.36 |
| 44,000 | 28.01 | 17.60 | 6.48 | 1.47 | 3.70 | 24.78 | 13.67 |
| 46,000 | 30.87 | 22.45 | 5.56 | 1.07 | 3.79 | 25.00 | 14.79 |
| 48,000 | 26.86 | 16.10 | 5.09 | 0.80 | 3.43 | 23.67 | 12.66 |
| 50,000 | 31.58 | 16.61 | 6.02 | 1.60 | 5.11 | 25.00 | 14.32 |

Beyond final performance, the dynamics of model improvement during training are also crucial. Table 7 details the accuracy of the 1B main model on six major downstream tasks during DRAW pretraining, with the learning curves further illustrated in Figure 6. We observe that DRAW consistently accelerates accuracy improvements across tasks, often reaching higher accuracies with fewer training steps compared to

the baseline. Detailed trend analysis also reveals that DRAW stabilizes early-stage training and mitigates performance volatility, highlighting its efficacy in guiding more efficient and stable learning.

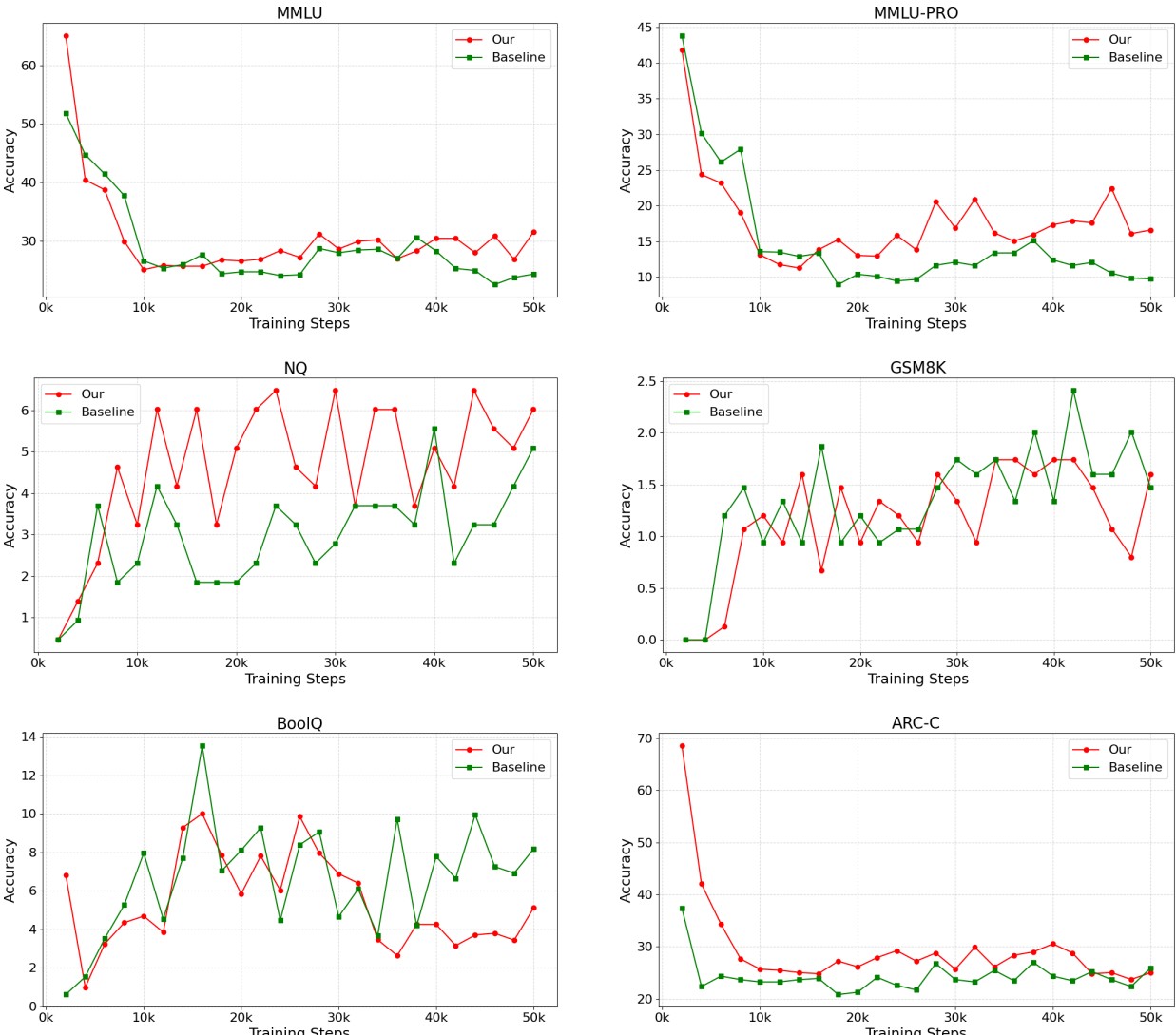

Figure 6: Accuracy(%) curves of six benchmarks (MMLU, MMLU-PRO, NQ, GSM8K, BoolQ, ARC-C) comparing DRAW and Baseline.

The stability of learned domain weights under stochastic initialization is analyzed by comparing the optimized weights across different random seeds for 17 domains, as shown in Table 8. While some local variances are observed, the overall weighting trends remain consistent, evidencing the robustness of the DRAW framework. This outcome suggests that even for rare or long-tail domains, the adaptive mechanism reliably allocates meaningful weights, which is crucial for real-world deployment.

Lastly, Table 7 systematically explores the effect of proxy model capacity on the learned domain weights. We compare results using proxy models of various sizes (70M, 150M, 510M, 280M). It's found that although a larger proxy can provide finer adjustments, the overall weighting trend remains essentially invariant across model scales. This indicates that DRAW's core mechanism does not depend on excessively large proxy models, supporting practical scalability and engineering feasibility.

In summary, this appendix systematically demonstrates the advantages of DRAW in terms of weight evolution, domain adaptability, and the robustness of the learned mixture—strongly supporting the main con-

Table 7: Domain weights learned by different proxy model sizes (70M, 150M, 510M, 280M) for DRAW.

| Domain | 70M | 150M | 510M | 280M |
|---|---|---|---|---|
| ArXiv | 0.0010 | 0.9709 | 0.7306 | 0.4735 |
| DM Mathematics | 0.0086 | 0.0027 | 0.0005 | 0.0004 |
| Enron Emails | 0.2933 | 0.0002 | 0.0002 | 0.0002 |
| EuroParl | 0.0004 | 0.0001 | 0.0002 | 0.0002 |
| FreeLaw | 0.0008 | 0.0003 | 0.0005 | 0.0006 |
| Github | 0.0225 | 0.0219 | 0.0719 | 0.0766 |
| Gutenberg (PG-19) | 0.0002 | 0.0002 | 0.0003 | 0.0003 |
| HackerNews | 0.0001 | 0.0002 | 0.0003 | 0.0003 |
| NIH ExPorter | 0.0002 | 0.0002 | 0.0003 | 0.0004 |
| PhilPapers | 0.0002 | 0.0002 | 0.0002 | 0.0003 |
| Pile-CC | 0.0003 | 0.0003 | 0.0007 | 0.0010 |
| PubMed Abstracts | 0.0007 | 0.0008 | 0.1904 | 0.4398 |
| PubMed Central | 0.0244 | 0.0004 | 0.0006 | 0.0007 |
| StackExchange | 0.2395 | 0.0004 | 0.0006 | 0.0007 |
| USPTO Backgrounds | 0.0003 | 0.0003 | 0.0004 | 0.0005 |
| Ubuntu IRC | 0.0002 | 0.0002 | 0.0002 | 0.0002 |
| Wikipedia (en) | 0.4075 | 0.0006 | 0.0021 | 0.0043 |

clusion that DRAW enables more effective and adaptable language model pre-training on large-scale, multi-domain datasets.

Table 8: Domain weights across 17 data domains under varying random seeds for DRAW.

| Domain | Seed = 1 | Seed = 115 | Seed = 123 | Seed = 17 | Seed = 42 |
|---|---|---|---|---|---|
| ArXiv | 0.0010150 | 0.0010100 | 0.0009531 | 0.0009537 | 0.0009646 |
| DM Mathematics | 0.0085540 | 0.0085540 | 0.0085540 | 0.0085550 | 0.0085530 |
| Enron Emails | 0.2933000 | 0.3295000 | 0.3036000 | 0.2641000 | 0.2770000 |
| EuroParl | 0.0003762 | 0.0003897 | 0.0003868 | 0.0003912 | 0.0003880 |
| FreeLaw | 0.0008116 | 0.0008223 | 0.0008085 | 0.0008287 | 0.0008117 |
| Github | 0.0225100 | 0.0215800 | 0.0219400 | 0.0216300 | 0.0213500 |
| Gutenberg (PG-19) | 0.0001532 | 0.0001557 | 0.0001554 | 0.0001563 | 0.0001562 |
| HackerNews | 0.0001374 | 0.0001394 | 0.0001391 | 0.0001400 | 0.0001396 |
| NIH ExPorter | 0.0002121 | 0.0002123 | 0.0002127 | 0.0002132 | 0.0002123 |
| PhilPapers | 0.0001590 | 0.0001594 | 0.0001592 | 0.0001594 | 0.0001588 |
| Pile-CC | 0.0002578 | 0.0002607 | 0.0002607 | 0.0002605 | 0.0002619 |
| PubMed Abstracts | 0.0006736 | 0.0006458 | 0.0006520 | 0.0006363 | 0.0006345 |
| PubMed Central | 0.0244400 | 0.0247600 | 0.0250200 | 0.0237100 | 0.0241000 |
| StackExchange | 0.2395000 | 0.2262000 | 0.2261000 | 0.2410000 | 0.2477000 |
| USPTO Backgrounds | 0.0002996 | 0.0002919 | 0.0002947 | 0.0002926 | 0.0002920 |
| Ubuntu IRC | 0.0001835 | 0.0001799 | 0.0001801 | 0.0001822 | 0.0001787 |
| Wikipedia (en) | 0.4075000 | 0.3852000 | 0.4106000 | 0.4368000 | 0.4171000 |

