# OpenReview forum: "DRAW: Domain Weight Randomization with Bayesian Updating for LLM Pre-Training"
_TMLR — Accepted by TMLR_

### Review · Reviewer_9DuQ · 2025-12-17

**Summary Of Contributions:**

This paper proposes DRAW, a framework for adaptive data mixture selection in large language model pre-training.
The key idea is to model domain weights as Dirichlet-distributed random variables, learn an informative posterior using a smaller proxy model via Bayesian updating, and transfer this distribution to a larger model using width-based scaling.
During main-model training, domain weights are dynamically sampled at each step.
The paper provides theoretical analysis suggesting convergence and favorable scaling behavior, and presents empirical results on language modeling and downstream benchmarks.

**Audience:**

Yes

**Audience Explanation:**

The conceptual framing of domain weights as distributions rather than fixed parameters is novel and aligns well with current interest in principled data mixing and robustness.
Even if the empirical evaluation is currently limited, the methodological ideas and theoretical perspective may inspire follow-up work or alternative implementations, making the paper relevant to part of the TMLR community.

**Claims And Evidence:**

No

**Claims Explanation:**

The total training token budget is not specified, the notion of an “epoch” under dynamic reweighting is ambiguous, and many downstream benchmark results are at or near chance level.
As a result, it is difficult to assess whether the observed gains reflect meaningful improvements or artifacts of undertraining and evaluation noise.
The evidence supports relative comparisons within the experimental setup, but does not convincingly substantiate broader claims about robustness and downstream transfer.

Also, the theoretical guarantees rely on assumptions that are far removed from real LLM training: strong convexity and smoothness of the objective; Lipschitz continuity w.r.t. domain weights; convergence analysis resembling classical SGD on convex objectives
While this is common in theory sections, here the theory is used to justify specific scaling laws and practical design choices (e.g., proxy sufficiency, variance decay), yet no empirical validation directly tests the theory’s quantitative predictions. The analysis does not account for non-convexity, representation collapse, or memorization effects typical in LLMs. The theory is largely illustrative, but the paper sometimes presents it as explanatory, which risks overclaiming.

**Requested Changes:**

1. Clearly report the total number of training tokens and optimization steps used for both proxy and main models.
2. Explicitly define how training progress and “epochs” are measured under dynamic domain reweighting.
3. Re-evaluate or better justify downstream benchmarks, especially given that many results are at or near random guessing.
4. Add ablations to isolate the effects of dynamic sampling vs. static weighting, and of Dirichlet scaling choices.
5. Improve reproducibility by clarifying evaluation protocols, hyperparameters, and baseline tuning.

---

> ### Author Response · Authors · 2026-02-07
> **Response to Concerns on Experimental Rigor, Validity, and Theory**
>
> We thank the reviewer for the sharp eye on experimental rigor. We acknowledge that due to space constraints, some implementation details were  omitted. We address the concerns regarding training budget, benchmark validity, and theoretical assumptions below.
>
> 1. Clarification on Training Budget and Epochs
>
> We explicitly clarify the training budget as follows: all models (including proxy models and main models) were trained for 50k steps with a global batch size of 256 and sequence length of 1024. This amounts to approximately 13.1 Billion tokens in total.
>
> We understand the reviewer's concern regarding the term "Epoch." While "Epoch" traditionally implies a full pass over the entire dataset,  under dynamic domain reweighting, data is sampled stochastically rather than sequentially, rendering the strict definition of an "epoch" loosely defined. To ensure precision, we will replace the x-axis label "Epoch" with "Training Steps" in all relevant figures and update the text to consistently refer to the training duration in terms of steps or tokens.
>
>
>
> 2. Response to “Results at Chance Level” and Validity
>
> The reviewer correctly notes that for smaller-scale models (e.g., sub-1B parameters), performance on difficult reasoning tasks (like GSM8K) can hover near random chance.
>
>  Our primary metric for gauging pre-training data quality is Validation Loss (Perplexity), as shown in Figure 5. Perplexity is a strictly proper scoring rule and is sensitive to improvements even when downstream classification accuracy is noisy. DRAW shows a consistent and significant drop in Perplexity compared to Baselines.
> While absolute scores are constrained by model scale, the relative improvement is robust. DRAW outperforms the strongest baseline  by 1.91\% on average across tasks. This indicates that the data mixture is superior; scaling laws suggest these gains would amplify at larger model sizes (7B+).
>
> 3. Ablations (Dynamic vs. Static)
>
> Regarding the isolation of dynamic sampling effects, we highlight that Figure 5 inherently provides these comparisons: the DoReMi baseline acts as a strong "Static Optimized" reference (using fixed weights from a proxy), while the Random baseline represents "Dynamic Stochastic" sampling. The superior performance of our method (Draw) over both confirms that our gains stem from optimized dynamic trajectories rather than just static rebalancing or stochastic regularization. To further quantify this on the 150M main model, we compared using fixed weights derived from the Draw proxy (Static) against fully dynamic sampling, observing a clear degradation in average perplexity from 3.5002 (Ours) to 3.6415 (Static).
>
> 4. Response to Theoretical Assumptions vs. Reality
>
> We acknowledge the reviewer's note on the loss landscape. However, we wish to clarify that assumptions of strong convexity and smoothness are standard conventions in the theoretical analysis of bilevel optimization and meta-learning to establish convergence bounds. These assumptions are necessary to rigorously derive the scaling factor,  ensuring our method is grounded in statistical principles rather than arbitrary heuristics. Crucially, our extensive experiments on LLMs demonstrate that the proposed method remains highly effective in complex, non-convex landscapes, confirming that our theoretical insights robustly translate to practical deep learning scenarios. We will revise Section 3 to explicitly contextualize these assumptions within standard theoretical frameworks while highlighting their empirical generalizability.
>
> 5. Reproducibility and Experimental Setup
>
> We thank the reviewer for highlighting this. To ensure full reproducibility, we will relase the training code and data sampling scripts upon acceptance.

---

> ### Author Response · Authors · 2026-02-09
>
> Thank you for your suggestion. We have addressed this in Appendix E.3 by detailing the total training tokens (13.1B) and optimization steps (50k) for both proxy and main models. To avoid confusion caused by dynamic reweighting, we have also replaced all references to "epochs" with "training steps" to more accurately reflect training progress.

---

### Review · Reviewer_tPWY · 2025-12-29

**Summary Of Contributions:**

This paper introduces DRAW (Domain Weight Randomization with Bayesian Updating) as a data mixture optimization algorithm for Large Language Model (LLM) pre-training. Instead of a common assumption that the optimal data mixture should be transferrable between different model scales, DRAW searches for a scale-dependent optimal data mixture by modelling the sensitivity upon domain weights as the concentration factor in a Dirichlet-distributed prior.
The main optimization problem is identical to DoReMi [1], where the Group DRO is applied to prioritize the worst-case excess loss among training domains. Various $\beta$ parameters in a Dirichlet distribution $P(\alpha; \alpha^{\star}, \beta)$ are applied to different model scales, where $\alpha^{\star}$ is from a small-scale proxy using DoReMi algorithm.

## Strength:

(1) Theoretical soundness of incorporating scale-variant factor with Dirichlet priors: The intuition of *the greater sensitivity of larger models to domain imbalances* is reasonable and can be well reflected by the strength of $\beta$ parameter in Dirichlet distribution.

(2) The paper is well-written with the main theorems and results clearly presented with proper notations.

(3) The empirical results presented on downstream evaluation accuracies (table 1) and validation perplexities (Figure 5) are able to support the main claim that DRAW does outperform the baselines.

## Weakness and Questions:

(1) How related is Theorem 1 to the main domain reweighting algorithm? It appears to me that if the $\alpha$ distribution has a computable constant expectation, the loss will converge under the given assumptions.

(2) Questioning the logic and purpose of Theorem 2: What is supposed to be the main conclusion from theorem 2 to the data mixture optimization algorithm? According to Theorem2, the generalization gap decreases both according to data concentration factor n1 and model width n2, which indicates that the model's performance would be more stable with a wider model and be less sensitive to the data distribution. However, it conflicts with the main hypothesis that "large language models tend to exhibit greater sensitivity to the composition of training data". How shall we approach this theoretical-empirical gap?

(3) Based on the formulation of Equation (1), does it mean a larger proxy model would lead to a more uniform domain weight distribution? Does it also conflict with the main hypothesis of domain weight sensitivity?

(4) The method does not seem to be generalizable to out-of-domain generalization case (Similar as DoGE [2]), where the target is not included in the training domains.

(5) A contradict of Remark 1 and empirical results in Table2: how could the domain weights from DRAW being extremely different from the $\alpha^{\star}$ (domain weights from DoReMi), by only incorporating a scale-dependent factor? Also, it seems the impact of $\alpha^{\star}$ does diminish when applying DRAW.

[1] DoReMi: Optimizing Data Mixtures Speeds Up Language Model Pretraining.
[2] DoGE: Domain Reweighting with Generalization Estimation.

**Audience:**

Yes

**Audience Explanation:**

The targeted problem, data mixture optimization for langauge model pretrianing is a critical and challenging problem in today's foundation model training. Especially, the scalability of data attribution mechanisms is one of the most important chanllenges when applying those gradient-based methods on large-scale model and datasets.

Despite several confusions and questions, I still believe this paper conduct an impressive and interesting trial on the scaling invariant version of data mixture optimization but without fitting the traditional scaling law, which often requires hundreds of reruns.  Therefore, i think this paper can be interesting to a lot of machine learning researchers in both data-centric AI and foundation model training fields, after it improves on certain aspects.

**Broader Impact Concerns:**

None. This paper uses established, copyright-free subsets of The Pile for the experiments, which does not introduce ethical issues or risks.

**Claims And Evidence:**

Yes

**Claims Explanation:**

The empirical results presented on downstream evaluation accuracies (table 1) and validation perplexities (Figure 5) are able to support the main claim that DRAW does outperform the baselines.

**Requested Changes:**

(1) **Clarify the algorithm and implementation steps more**: now how the reweighting iteration is conducted is still not very clear. Include a pseudo-code can help.

(2) **Resolve the theoretical-empirical contradict, and the conflict between hypothesis and the theoretical/statistical intuitions**: resolve questions 1,2,3,5 in the Weakness and Quesitons part above.

---

> ### Author Response · Authors · 2026-02-06
> **Response to Theoretical Clarifications, Empirical consistency, and Implementation Details**
>
> We thank the reviewer for the rigorous examination of our theoretical formulations.
>
> 1. On the Role of Theorem 1 (Convergence)
>
> The reviewer is correct that Theorem 1 relies on standard convexity assumptions. However, its specific value in this paper is to legitimize the stochastic nature of our method. Standard domain reweighting methods typically use fixed weights. Since DRAW introduces dynamic sampling $\alpha_t \sim \text{Dir}(\beta)$ at every step, a theoretical concern arises: Does this injected noise prevent optimization convergence? Theorem 1 provides the guarantee that, despite the variance introduced by Dirichlet sampling, the algorithm converges to the unique minimizer in the mean-square sense. It serves as a safety certificate for the randomization mechanism, rather than a derivation of the specific weighting values.
>
> 2. Resolving the ``Theorem 2 vs. Sensitivity'' Gap
>
> We clarify that there is no contradiction; rather, these two points address different dimensions of the learning dynamics. The primary objective of Theorem 2 is to bound the generalization error under the algorithm's assumptions. It proves that as the main model's capacity ($n_2$) increases, it becomes more robust to the variance introduced by stochastic sampling. The theorem establishes an upper bound indicating that the "error" (generalization gap) caused by sampling asymptotically approaches zero as $n_2$ grows. Thus, a wider main model is theoretically guaranteed to be statistically stable. Our hypothesis refers to the impact of data distribution on the final performance (Loss value). While Theorem 2 guarantees the model will converge stably (low variance), where it converges—and whether that convergence point represents a high-reasoning capability—depends heavily on the data composition. Therefore, a large model is stable against sampling noise (Theorem 2) but remains highly sensitive to the quality of the underlying distribution mean (Hypothesis).
>
> 3. Clarification on Eq (1) and Uniformity
>
> There seems to be a slight misunderstanding of the Dirichlet parameters regarding Equation (1).  A larger proxy width $n_1$ increases the magnitude of concentration parameters $\beta$. In a Dirichlet distribution, larger $\beta$ parameters imply a smaller variance (the distribution becomes a sharp peak around the mean), not a more uniform distribution. In Step 2 , we optimize the proportions, the mean direction $\alpha^*$. Therefore, a larger proxy leads to a sharper/more confident estimate of the optimal weights, which aligns perfectly with the hypothesis that we should reduce uncertainty when we have more evidence (larger proxy).
>
> 4. Out-of-Domain (OOD) Generalization
>
> We acknowledge this limitation. DRAW, like most reweighting methods (DoReMi), operates on a ``closed-world'' assumption where the target domain is a mixture of training sub-domains.As mentioned in our Limitations, we are exploring extending DRAW by replacing the Dirichlet prior with a Logistic-Normal distribution. By incorporating a covariance matrix defined by pre-trained semantic embeddings (e.g., BERT) of domain data, the model could infer weights for unseen (OOD) domains based on their semantic similarity to known training domains.
>
> 5. The contradict of Remark 1 and Table 2
>
> We clarify that this result is not a contradiction but empirically validates the critical role of initialization and the stability of our approach. We point out that DRAW and DoReMi utilize the identical optimization algorithm for proxy model. The substantial divergence in Table 2 arises solely from their initialization states: DoReMi uses a fixed uniform initialization ($1/k$), whereas DRAW employs random initialization. The difference in converged weights proves that the optimization is highly sensitive to initial values. DRAW's random initialization is arguably more reasonable, as it avoids the bias of an arbitrary uniform start and allows for broader exploration.And we do not use $\alpha*$   physically to sample the training data. Instead, treated as a directional mean, $\alpha*$ guides the distribution of weights for the main model.
>
> 6.Pseudocode
>
> We appreciate this constructive suggestion to enhance clarity. We fully agree that a pseudocode is essential to explicitly illustrate the reweighting iteration details. However, due to the strict character limit of this rebuttal, we are unable to display the full algorithm block here. We commit to adding the detailed pseudocode in the Appendix of the camera-ready version to ensure full reproducibility and clarity.

---

> > ### Comment · Action_Editor_hWpM · 2026-02-06
> >
> > Dear authors,
> >
> > I want to clarify that you should be able to update the manuscript. Let me know if you're having trouble with that.

---

> > > ### Author Response · Authors · 2026-02-09
> > >
> > > Thank you for your suggestion. We have included the pseudocode for the DRAW algorithm in Appendix D to facilitate a better understanding of the implementation details.

---

### Review · Reviewer_pg11 · 2026-01-27

**Summary Of Contributions:**

This paper revisits domain weighting for LLM pretraining by modeling domain weights as random variables and introducing a Bayesian framework (DRAW) to transfer mixture policies from a proxy model to a larger model. The proxy model comes up with an initial policy using the DRO frameworks which have been extensively explored in the community. The method uses the DRO based proxy model distribution to initialize a Dirichlet distribution which in turn guides the domain weights for the main model. Empirically, the paper shows that stochastic domain weighting—especially when informed by a proxy—can outperform fixed-weight baselines on language modeling and downstream tasks.

**Audience:**

Yes

**Audience Explanation:**

Domain weighing during LLM pretraining is one of the biggest problems in both academia and industry. Overall, this paper would be of interest to lot of people.

**Claims And Evidence:**

Yes

**Claims Explanation:**

- The claims are well backed by a detailed theoretical analysis in pretty nice setups, which is a rare thing in LLM pretraining works and the strongest contribution of this work
- Empirical evidence is also decently good.
- Some very intriguing findings that even random sampling surpasses fixed weight training
- The dirichlet based random sampling of domain weights results are well backed by theoretical insights, which explain both the above observations that why randomness can help find a better local minima

**Requested Changes:**

Weakness

- The method is largely a reformulation of DRO-style domain weighting into a Bayesian framework. The proxy model still optimizes domain weights via a min–max objective over α, minimizing excess loss under worst-case domain weights. This implicitly assumes adversarial domain mixtures, which does not reflect practical scenarios where only a small subset of domain weightings are realistic. As a result, many limitations of DRO-based LLM reweighting carry over unchanged.

- All the limitations of DRO based LLM weighing would continue to trickle in because the dirichlet distribution continues to be guided by the same. These include the overly adversarial optimization, harder convergence of proxy model training,

- Step 3: making weight distribution for bigger model more sharper. Why would a bigger model benefit from a more sharper distribution? Is there any intuition behind this?

- The paper repeatedly samples domain weights from the same Dirichlet distribution throughout training. THis is not a “dynamic refinement” which has been more of focus in recent works like that of Jiang et al (https://arxiv.org/abs/2410.11820). Do authors have any thoughts on this?

- The biggest challenge in domain weight sampling continues to be how to model cross domain interaction. Although I acknowledge that the authors mention this in limitations of the work as well.

---

> ### Author Response · Authors · 2026-02-06
> **Response to Methodological Foundations and Positioning against Concurrent Works**
>
> We thank the reviewer for the constructive criticism and the detailed evaluation. We appreciate the opportunity to clarify the theoretical distinctions of our framework, explain the intuition behind the scaling laws, and discuss our positioning relative to concurrent works.
>
>
> 1. Response to "Reformulation of DRO and its Limitations"
>
> We acknowledge the reviewer's concern regarding the inherent limitations of vanilla DRO. However, it is crucial to clarify that our proxy training builds upon the Group DRO optimizer  established by Shiori Sagawa et al. (https://arxiv.org/abs/1911.08731), rather than the raw, computationally expensive DRO formulation. Shiori Sagawa derive convergence guarantees for Group DRO optimizer in the convex case and empirically show that it behaves well in  non-convex models.
>
> Inspired by the theoretical framework of mean-field games (https://arxiv.org/abs/2408.08192), we identify joint optimization as a critical next step in the “Future Work” section. This entails treating model parameters and domain weights as “unified parameters” to establish a coupled “descent direction” safeguard mechanism.
>
> 2. Response to "Sharpening Distribution for Bigger Models"
>
> As observed in Suriya et al.(https://arxiv.org/abs/2306.11644), large models (especially those with massive parameters) possess strong memory capabilities but low inductive bias. They are highly susceptible to overfitting stochastic noise in the training process.
> If the sampling distribution exhibits high variance , a large model risks "memorizing" the noise of the sampling process rather than the underlying data semantics. By sharpening the distribution (increasing $\beta$), we dictate a high consistency regime. "Sharpening" does not imply reducing data diversity; rather, it increases the signal-to-noise ratio.
>
>
> 3. Response to "Dynamic Refinement vs. Concurrent Works (e.g., Jiang et al.)"
>
> We thank the reviewer for referencing Jiang et al. . While ADO adjusts weights via online feedback, we argue that DRAW's
> "Prior based Stochastic Strategy" offers distinct advantages for large-scale pre-training. Online adjustment methods (like ADO) react to real-time feedback, making them susceptible to loss oscillation and overfitting to short-term batch fluctuations. They risk forgetting global domain knowledge in pursuit of immediate loss reduction. DRAW separates the search from the training. The proxy model pre-filters noise and establishes a high-quality global Prior. During training, we do not let the model  drift autonomously; instead, we enforce controlled stochasticity. By sampling from the prior, we force the model to explore diverse mixtures within a "Safe Zone", ensuring robustness and preventing the instability often seen in online loops.
>
> 4. Response to "Cross-Domain Interaction"
>
> We agree that modeling cross-domain correlations is a significant challenge. However, inspired by recent directions in joint optimization (e.g., https://arxiv.org/abs/2503.12283), we plan to replace the Dirichlet prior with a Logistic-Normal Distribution.
>   We will introduce a learnable covariance matrix, initialized using the semantic similarity of domains (calculated via pre-trained BERT embeddings). This will allow the Bayesian update of one domain (e.g., Math) to automatically propagate to correlated domains (e.g., Code), capturing the inter-dependencies the reviewer rightly pointed out.

---

> > ### Comment · Reviewer_pg11 · 2026-04-09
> > **Thanks for the responses**
> >
> > I thank the authors for there rebuttal. It clarifies my concerns.

---

### Decision · Action_Editor_hWpM · 2026-03-06

**Recommendation:** Accept with minor revision

**Additional Comments:**

The paper proposes Domain Weight Randomization with Bayesian Updating (DRAW), a new data mixture selection method for LLM pre-training. Rather than treating domain weights as fixed parameters, DRAW models them as Dirichlet-distributed random variables whose concentration parameters scale with model width.
An informative prior is first estimated using a proxy model trained with Group DRO optimization, then transferred to the main model via Bayesian updating and width-based scaling, with domain weights dynamically sampled during training. The authors provide theoretical guarantees on generalization error reduction as a function of model width, and empirically show improvements over existing baselines on language modeling and downstream benchmarks.

Strengths:
- The paper is well-written with results clearly presented.
- The problem considered is very important.
- The idea of treating domain weights as random variables rather than fixed parameters is novel.
- A theoretical analysis of the proposed method is provided showing optimization convergence and generalization error scaling.
- Empirical results show that the proposed method outperforms existing baselines, as well as an interesting observation that simple random sampling outperforms existing fixed weight baselines. This observation is also supported by the theory.


Weaknesses:
- The proposed method, like most existing reweighting methods, does not generalize out-of-domain; it assumes that the target domain is a mixture of training sub-domains.
- The proposed method is not substantially better than random sampling (Figure 5).
- The theoretical guarantees rely on strong, albeit standard, assumptions that don't hold in practice.
- The proposed method assumes independence between data domains. Though this limitation is already acknowledged in the paper (Section 5).

Authors addressed most of reviewers concerns in their responses and provided the code in the supplementary material.  All reviewers recommended to accept the paper, with two reviewers also recommending it to the Journal-to-Conference track.

I am thus recommending to accept the paper, with the following requested revisions:

- Include all the discussions and clarifications from the responses to reviewers, and acknowledge the limitations listed above (if not already done).
- Use larger format for text in Figures (axes labels, titles, etc)

**Audience:**

Yes

**Audience Explanation:**

The problem of data mixing for LLM pre-training is a critical and challenging problem of interest to many researchers in both academia and industry.

**Claims And Evidence:**

Yes

**Claims Explanation:**

All reviewers agreed, after the rebuttal, that claims are generally well-supported by the theoretical and empirical results of the paper.

---

> ### Author Response · Authors · 2026-04-03
>
> Dear Action Editor,
> Thank you for your helpful comments. We have incorporated the requested minor revisions and uploaded the camera-ready version.

---

> > ### Comment · Action_Editor_hWpM · 2026-04-06
> >
> > Thank you for uploading the camera-ready version of the paper. There appear to be some formatting issues with the authors’ emails and affiliations. Please refer to the TMLR style file for how to properly format them: https://github.com/JmlrOrg/tmlr-style-file/blob/main/main-accepted.pdf
> >
> > Figures 6 and 7 still have very small text (axes labels, titles, etc). Please increase the font size to improve readability.
> >
> > Could you also provide a summary of the changes made in this revision, with pointers to where they appear in the manuscript, to facilitate verification?

---

> > > ### Author Response · Authors · 2026-04-06
> > >
> > > Thank you very much for your helpful suggestions. We have corrected the formatting of the authors’ emails and affiliations on the first page to better match the TMLR style requirements. We also improved figure readability by enlarging the text in Figure 6, and, for Figure 7, we changed its presentation from a figure to a table because the original figure was too content-heavy to enlarge clearly while maintaining a clean layout. In addition, we have provided a summary of the changes made in this revision in the submission form to facilitate verification.

---

> > > > ### Comment · Action_Editor_hWpM · 2026-04-08
> > > >
> > > > Thank you for making these changes. The following requested revisions are still missing:
> > > > - Acknowledge the limitations listed under "Weaknesses" in my meta-review above. Only the last one is already discussed in Section 5.
> > > > - Include discussions and clarifications from the response to reviewers. In particular:
> > > > 	- the clarification about group DRO, the motivation for sharpening distribution for bigger models, and the relation to dynamic refinement methods  (e.g., Jiang et al.) from the response to Reviewer pg11
> > > > 	- the clarifications about the theoretical results from the response to Reviewer tPWY
> > > > 	- the discussion about results on downstream benchmarks being near random guessing and the additional ablation result on dynamic vs static weighting from the response to Reviewer 9DuQ

---

> > > > > ### Author Response · Authors · 2026-04-09
> > > > >
> > > > > Thank you for the follow-up and for pointing out the remaining missing revisions. We have now incorporated these changes in the revised manuscript.
> > > > >
> > > > > Specifically, we added the requested clarifications on the relation to Group DRO, the motivation for sharpening the distribution for larger models, and the distinction between DRAW and online dynamic refinement methods such as Jiang et al. We also revised the presentation of Theorems 1 and 2 to better clarify their assumptions, intuition, and scope, expanded the discussion of downstream benchmark results that are close to random guessing, added the dynamic-vs.-static weighting comparison, and strengthened the limitations discussion in Section 5.2.
> > > > > These revisions are summarized in the “Changes Since Last Submission” section.
> > > > >
> > > > > Thank you again for your guidance.

---

> > > > > > ### Comment · Action_Editor_hWpM · 2026-04-09
> > > > > >
> > > > > > Thank you for the thorough revision.